# MemFreezing: A Novel Adversarial Attack on Temporal Graph Neural Networks under Limited Future Knowledge

Yue Dai [* 1]   Liang Liu [* 2]   Xulong Tang [1]   Youtao Zhang [1]   Jun Yang [2]

## Abstract

Temporal graph neural networks (TGNN) have achieved significant momentum in many real-world dynamic graph tasks. While most existing TGNN attack methods assume worst-case scenarios where attackers have complete knowledge of the input graph, the assumption may not always hold in real-world situations, where attackers can, at best, access information about existing nodes and edges but not future ones after the attack. However, studying adversarial attacks under these constraints is crucial, as limited future knowledge can reveal TGNN vulnerabilities overlooked in idealized settings. Nevertheless, designing effective attacks in such scenarios is challenging: the evolving graph can weaken their impact and make it hard to affect unseen nodes. To address these challenges, we introduce MemFreezing, a novel adversarial attack framework that delivers long-lasting and spreading disruptions in TGNNs without requiring post-attack knowledge of the graph. MemFreezing strategically injects fake nodes or edges to push node memories into a stable "frozen state," reducing their responsiveness to subsequent graph changes and limiting their ability to convey meaningful information. As the graph evolves, these affected nodes maintain and propagate their frozen state through their neighbors. Experimental results show that MemFreezing persistently degrades TGNN performance across various tasks, offering a more enduring adversarial strategy under limited future knowledge.

*Equal contribution  [1]Department of Computer Science, University of Pittsburgh, Pittsburgh, USA [2]Department of Electrical and Computer Engineering, University of Pittsburgh, Pittsburgh, USA. Correspondence to: Yue Dai <yud42@pitt.edu>, Liang Liu <lil125@pitt.edu>.

*Proceedings of the 42$^{nd}$ International Conference on Machine Learning*, Vancouver, Canada. PMLR 267, 2025. Copyright 2025 by the author(s).

## 1. Introduction

Dynamic graphs are prevalent in real-world scenarios, spanning areas like social media (Kumar et al., 2018), knowledge graphs (Leblay & Chekol, 2018), autonomous systems (Leskovec et al., 2005), and traffic graphs (Pareja et al., 2020). Inspired by the success of GNNs (Kipf & Welling, 2016; Hamilton et al., 2017; Veličković et al., 2017; Xu et al., 2018), Temporal Graph Neural Networks (TGNNs) have become leading solutions for dynamic graph tasks (Trivedi et al., 2019; Kumar et al., 2019; Rossi et al., 2020; Zhang et al., 2023; You et al., 2022). As such, there is a pressing need to study their robustness towards adversarial attacks, especially since such attacks have shown significant efficacy against traditional GNNs (Wang et al., 2018; Tao et al., 2021; Zügner et al., 2018; Zou et al., 2021; Ma et al., 2020; Zang et al., 2020; Bojchevski & Günnemann, 2019; Sun et al., 2022; Li et al., 2022). By modifying the input graphs with imperceptible and subtle perturbations, the adversarial attacks can make the models yield incorrect or adversary-expected results. For instance, a social media such as REDDIT (Kumar et al., 2018) may employ TGNN to decide whether comments (as edges) from users to posts (as nodes) should be banned based on his/her comment histories. With subtle adversarial attacks, malicious messages can easily bypass this checking functionality.

While several studies have explored the effectiveness of adversarial attacks on dynamic graphs (Lee et al., 2024; Sharma et al., 2022; 2023; Chen et al., 2021), they often assume that attackers have complete knowledge of the input graphs at the time of the attack, which may be challenging to achieve in many real-world scenarios. In practice, as dynamic graphs evolve, by the time attackers observe the entire evolution (i.e., track all changing nodes and edges) and identify the optimal timestamps to inject adversarial perturbations (e.g., adding fake nodes or edges), those key timestamps may have already passed, making it challenging to inject noises timely. Hence, in real-world cases, despite white-box (i.e., model parameters are known) or black-box setups (i.e., model parameters are unknown), the adversary may attack TGNN without knowing future changes on the graph. Therefore, studying TGNN adversarial attacks under these real-world constraints is essential since attacking un-

der limited future knowledge may exhibit unique patterns that reveal TGNN vulnerabilities overlooked in idealized, full-knowledge analyses. However, attacking TGNNs with limited knowledge up to the attack time faces significant challenges due to the evolving nature of dynamic graphs. First, the impact of adversarial noise can quickly decay as the graph evolves and node information updates. Second, it is difficult to influence unseen nodes or edges that appear after the attack, as their information is unknown. Thus, an effective strategy must endure the graph's evolution and affect both current and future nodes despite this uncertainty.

Interestingly, the node updating mechanism in Temporal Graph Neural Networks (TGNNs) offers unique potentials for persisting and propagating adversarial noises in dynamic graphs. Generally, TGNNs maintain and update node status vectors, often referred to as *node memory* by recent studies (Rossi et al., 2020; Zhou et al., 2022; Wang & Mendis, 2024; Zhou et al., 2023; Wang & Mendis, 2023), to capture nodes' temporal history, which is crucial for delivering accurate predictions in dynamic graph tasks. Moreover, a node's memory vector can potentially affect its neighbors. When graph changes occur—such as the addition and deletion of nodes or edges—the memory vectors of related nodes are updated based on their neighbors' memories. This raises intriguing questions: *Can TGNN predictions be disrupted by disabling their memories, and can this effect persist and spread through their memory updates?*

To address this inquiry, we thoroughly investigated the memory update patterns of nodes within TGNNs and made the following observations: (1) Although it is not possible to directly affect unseen predictions, we can degrade TGNN prediction accuracy by pushing nodes—whether seen or unseen—into a relatively 'frozen' state, their memories remain stable and exhibit limited responsiveness to surrounding changes, reducing their ability to convey updated or meaningful information. (2) While a noisy node's memory vector may struggle to maintain its noisy state over time on its own, this state can persist for much longer if its neighboring nodes have similar memory states.

To this end, we introduce MemFreezing, a novel adversarial attack to expose TGNN vulnerabilities under limited knowledge about input graphs. At a specific attack timestamp, MemFreezing strategically selects groups of victim nodes that reinforce each other's noisy states using a scheme called *cross-freezing*. By injecting carefully crafted fake messages, MemFreezing leads nodes into a stable "frozen state", reducing their responsiveness to graph changes and thereby misleading predictions. Additionally, it simulates victim nodes' future neighbors to encourage the propagation of the frozen effect. We summarize our contributions as follows:

- We identify a highly possible threat model where attack-

ers only see the graph up to the attack time, posing unique challenges for adversarial attacks on TGNNs.

- We propose MemFreezing, which disrupts TGNN node memories by pushing them into unnaturally stable states. It adopts a cross-freezing mechanism to keep nodes' memories stable despite future updates and encourages affected nodes to propagate stable states by simulating their future neighbors.

- We compare our method with prior GNN adversarial attacks on various dynamic graphs. Experimental results show that, MemFreezing effectively and persistently misleads TGNN predictions across diverse datasets and models, outperforming state-of-the-art GNN attacks, even in the presence of defenses.

## 2. Background and Related Work

**Dynamic Graphs.** Unlike a static graph, a dynamic graph consists of nodes and edges evolving over time. Dynamic graphs can be represented in two ways: Discrete-Time Dynamic Graphs (DTDGs) describe dynamic graphs as a series of static snapshots taken periodically, while Continuous-Time Dynamic Graphs (CTDGs) view the graph as a collection of events—each event detailing updates like node or edge changes. Recent TGNNs focus on CTDGs since they can retain more information than DTDGs' fixed intervals and more complex (Kazemi et al., 2020). Within the CTDG paradigm, the dynamic graphs are represented as $G = \{x(t_1), x(t_2), ...\}$, in which $x(t_i)$ indicates an event happened at timestamp $t_i$. Generally, the prediction task for CTDGs can be depicted in equation 1.

$$y_i = f_\theta(G_i, t_i) = f_\theta(\{x(t_1), x(t_2), ...x(t_{i-1})\}, t_i) \quad (1)$$

At the prediction time $t_i$, the model $f_\theta(\cdot)$ takes all previous events $G_i = \{x(t_1), x(t_2), ...x(t_{i-1})\}$ as inputs and predicts the testing nodes' classes or future edges.

**Temporal Graph Neural Networks.** The memory-based Temporal Graph Neural Networks (TGNN) are widely studied and achieve state-of-the-art accuracies in dynamic graph tasks (Trivedi et al., 2019; Kumar et al., 2019; Rossi et al., 2020; Kazemi et al., 2020; Zhang et al., 2023; You et al., 2022; Ahmadi, 2020). Generally, these TGNNs maintain a state vector for each node that tracks the node's history, and use it for predictions. Note that, despite their different names (e.g., node memories (Rossi et al., 2020; Wang et al., 2021), node representations (Trivedi et al., 2019), node dynamic embeddings (Kumar et al., 2019)) across various TGNNs, these node features are represented as vectors on presented nodes and evolve over time to capture temporal information of these nodes. Following existing general TGNN frameworks (Zhou et al., 2022; 2023; Wang & Mendis, 2024; 2023; Rossi et al., 2020), we refer to these evolving node feature vectors as ***node memories***. As illus-

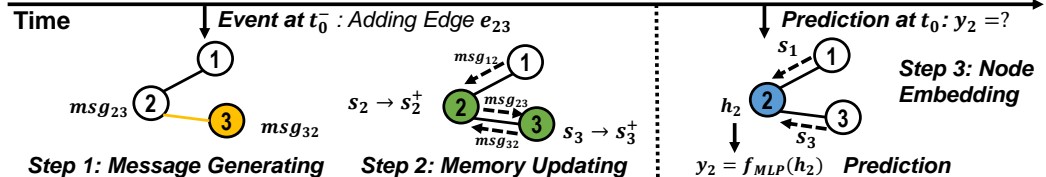

*Figure 1.* The three steps of TGNN computing assuming a new event at timestamp $t_0^-$ adds an edge $e_{23}$ to the dynamic graph: Firstly, messages $msg_{23}$ are generated for the nodes involved in this event nodes 2 and 3. Next, the nodes aggregate messages from their neighbors and update their memories (e.g., $s_2 \rightarrow s_2^+$). At a future prediction time $t_0$, nodes aggregate memories (e.g., $s_1$ and $s_2$) from their neighbors and embed them into node vectors (e.g., $h_2$) for the prediction.

trated in Figure 1, TGNNs produce node embedding for the predictions in three steps. When an event $x(t_i)$ adds an edge $e_{uv}$ from node $u$ to node $v$ (i.e., $x(t_i) = e_{uv}$), two messages are generated as equation 2 (***Step 1***). For simplicity, we only present the updating and following operations of node $u$, which is the same for node $v$.

$$m_{vu} = msg(s_v, s_u, \Delta T, e_{uv}) \qquad (2)$$

The $msg(\cdot)$ is a learnable function such as Multi-Layer-Perceptions (MLPs). The $s_u$ and $s_v$ denote the memories of node $u$ and node $v$ at their last updated times, and $\Delta T$ represents the difference between the current timestamp and the nodes' last updated times. Next, nodes $u$ and $v$ aggregate messages from their neighbors and update their memories as equation 3 (***Step 2***).

$$s_u^+ = UPDT(s_u, AGGR(m_{ku}|k \in N(u))), \qquad (3)$$

The $N(u)$ denotes the neighbors of node $u$. The $AGGR(\cdot)$ is usually implemented by a $mean$ or $most\_recent$ function to aggregate messages from the node's neighbors (Rossi et al., 2020). The $UPDT(\cdot)$ uses the aggregated messages to update the node's memory and is usually implemented by a Gated-Recurrent-Unit (GRU) (Chung et al., 2014). When there is a prediction involving node $u$, TGNNs use a graph embedding module, such as Graph Attention Network (Veličković et al., 2017), to embed the node's memory into the final node embedding, as depicted in equation 4 (***Step 3***).

$$h_u = GNN(s_u, s_k|k \in N(u)), \qquad (4)$$

During prediction, TGNNs use nodes' latest memories to compute the node embedding $h_u$. The resulting node embedding $h_u$ is fed into an MLP for the final predictions.

**Adversarial Attacks on Graph Neural Networks.** The considerable achievements of GNNs have catalyzed numerous investigations into their resilience against adversarial attacks (Chen et al., 2017; Bai et al., 2018; Wang et al., 2018; Zügner et al., 2018; Bojchevski & Günnemann, 2019; Ma et al., 2020; Zang et al., 2020; Tao et al., 2021; Zou et al., 2021; Sun et al., 2022; Li et al., 2022; Zou et al., 2023). These adversarial attacks generally seek to misguide

GNN predictions by modifying the nodes and edges of input graphs. For example, (Wang et al., 2018) introduces fake nodes with fake features that can minimize the loss between prediction results in the original graphs and the targeted fake results; (Zügner et al., 2020) adds and deletes edges that can cause the most substantial increases in the training losses on the original graphs. Recently, there have also been a few studies that explored the effectiveness of adversarial attacks on dynamic graphs and TGNNs (Lee et al., 2024; Sharma et al., 2023; 2022; Chen et al., 2021).

## 3. Problem Analysis

**Threat Model under Limited Future Knowledge.** Prior TGNN attacks (Lee et al., 2024; Sharma et al., 2023; 2022; Chen et al., 2021) assume that attackers have full knowledge of the target graphs and that these graphs remain static after the attacks. However, this assumption may not hold in many real-world settings, as attackers cannot return to the optimal attack times after observing the entire evolution of a dynamic graph. In particular, when an attacker observes the evolution of a dynamic graph at $t_n$ and identifies optimal attack timestamps $t_{a_1}, t_{a_2}, ..., t_{a_k} \leq t_n$, they would need to go back to these past timestamps to inject noise, which is infeasible in practice. To this end, we assume that an attacker's knowledge is limited to events up to the attack timestamp, and the graph continues to evolve afterward. Specifically, we set up the attack model as follows.

- *Attacker's Goal:* Given an evolving dynamic graph and a TGNN model, the attacker's goal is to misguide the TGNN predictions by introducing a limited amount of changes to the entire graph (e.g., affecting a small number of total nodes limited by the attack budget.)
- *Attacker's Knowledge:* Attackers have access only to information up to the attack's timestamps—namely, model parameters, presented graph inputs, and node memories before the attack—but not future graph changes. Regarding acquiring presented graph information, platforms like Wikipedia, Reddit, Meta, or X typically offer publicly accessible dynamic graphs, allowing adversaries to reconstruct them reasonably. Likewise, many TGNN architectures and pre-trained models are open-sourced, and

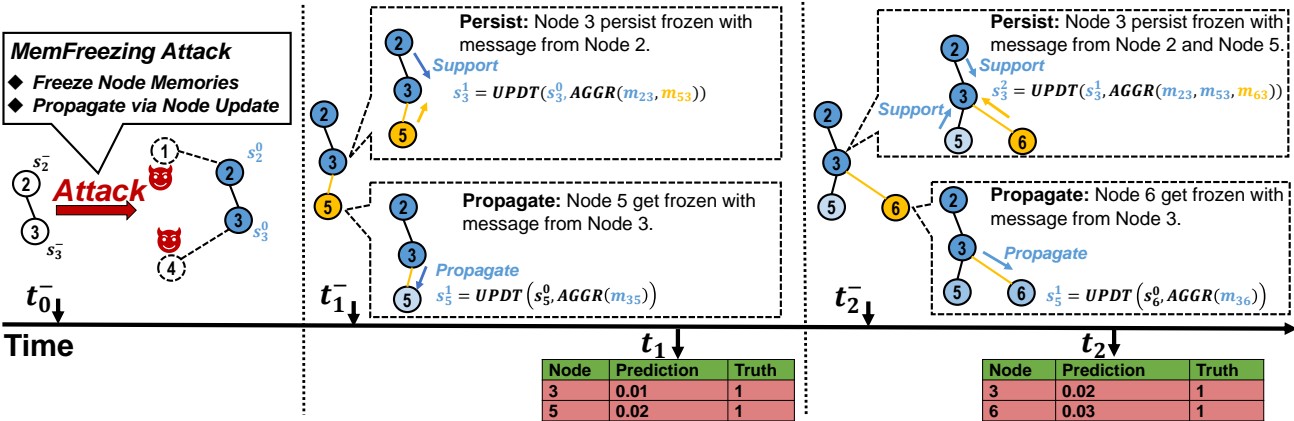

*Figure 2.* The resultant adversarial effects of the MemFreezing attack. (1) Victim nodes are kept frozen with support (i.e., similar messages) from their frozen neighbors, such as node 3, which is kept frozen at $t_1^-$ and $t_2^-$. (2) Victim nodes propagate frozen states to their future neighbors by inducing them to be similar, such as node 3 propagates to node 5 at $t_1^-$ and 6 at $t_2^-$.

even when they are not, techniques like insider threats or model extraction (Yao et al., 2024; Oliynyk et al., 2023) can be used to obtain model parameters. (We also discuss black-box scenarios in Appendix B.15.)

- *Attacker's Capability:* Attackers can add fake events as adding nodes/edges at the attack time. For example, while attacking TGNNs in social media, attackers can create fake user accounts as fake nodes and make junk comments to the blogs as fake edges.

**Challenges in Limited Future Knowledge.** Due to limited knowledge up to the attack time, adversarial attacks on TGNNs must contend with unknown future changes in dynamic graphs. However, the subsequent changes after the attack may significantly limit the attack performances for two reasons: First, for seen and attacked targets, the noise that misleads their predictions becomes mixed with new information from future changes (as described in equation 3 and equation 4), making it too weak to mislead future predictions. Second, unseen nodes and edges added after the attack are difficult to affect, as the attackers have no knowledge of these future elements and cannot generate effective noise to mislead them. As details shown in Section 5 (e.g., Figure 5 and Table 1), while existing GNN attacks (Wang et al., 2018; Zou et al., 2021; Li et al., 2022) effectively reduce the model's accuracy immediately after the attack, they struggle to perturb predictions in the future timestamps.

## 4. The MemFreezing Attack

We propose MemFreezing, an adversarial attack specifically tailored for TGNNs. It consists of two key features: i) To create long-lasting adversarial effects, we induce nodes to mutually lock their memories, keeping them stable during future updates. Consequently, the victim nodes become less responsive to surrounding changes, limiting their ability to

provide critical information for predictions. ii) To affect unseen nodes and edges, we simulate future neighbors for the victim nodes and encourage these victim nodes to update the memories of their simulated future neighbors into similar, stable states. As a result, the adversarial effects remain persistent through future changes and influence subsequent predictions, as illustrated in Figure 2.

**Memory Freezing Objective.** Instead of focusing on maximizing prediction losses as prior adversarial attacks, which are limited by unknown and diverse future events, we propose to transform victim nodes' memories into similar and stable states, which we refer to as ***Frozen State***. In particular, by keeping node memories similar and unchanged over time, nodes in TGNNs can hardly carry or convey meaningful information, consequently disturbing predictions. To quantitatively investigate the potential effectiveness of freezing node memories, we freeze the node memories in TGN (Rossi et al., 2020) and JODIE (Kumar et al., 2019) by consistently forcing their node memories to all zero, then evaluate their performances on edge prediction tasks on Wikipedia (Kumar et al., 2018) dataset. As shown in Figure 3(a), this leads to significant accuracy drops over time, demonstrating the impact of freezing node memories. Thus, we set our attack objective as freezing node memories unchanged.

### 4.1. Freezing and Persisting Node Memory

**Challenges in Persisting Frozen Memories.** However, keeping nodes' memories in frozen states is challenging, as unpredictable neighbor messages can significantly alter a node's memory during updates. An intuitive solution is to minimize the impact of incoming messages during an update. To explore the feasibility of this approach, we conducted a case study using TGN on the Wikipedia dataset. Specifically, we sampled 100 victim nodes, manipulated each node's memory to minimize the interference of messages during

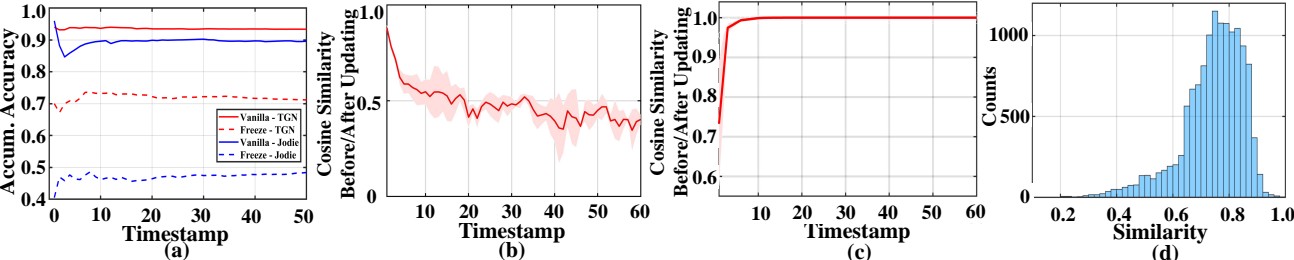

*Figure 3.* (a) The accumulated accuracy in vanilla TGNNs and their frozen counterpart. The ranges (colored bar) and averages (line) of the cosine similarities between node's evolving memories with (b) *persisting frozen by node themselves* and (c) *by connected groups (cross-freezing)*. (c) The distribution of cosine similarities among the ideal frozen states in different nodes.

updates, and assessed whether they remained unchanged over subsequent timestamps. As shown in Figure 3(b), the cosine similarities between nodes' pre- and post-update memories still dropped remarkably, indicating that future messages cannot be fully blocked in RNN or attention-based memory updating modules. We show further experimental details and theoretical analysis in Appendix A.1.

**Opportunities in Freezing.** Although blocking messages from a node's neighbors is infeasible, we can inject noise into those neighbors so that their updates help sustain the node's frozen memory. In particular, we assume that *node memories in TGNN can remain stable when surrounded by neighbors with similar memories.* We verify the assumption using the same model and data as Figure 3(b). Specifically, we first sample one-third of 100 victim nodes as *root node*, then sample two neighbors for each root node (referred to as *support neighbors*) and set their memories the same as the root node, then observe their memory changes over time. As depicted in Figure 3(c), if nodes have similar neighbors, their memories quickly converge to a relatively stable state and persist through future changes—we term this state as the node's *ideal frozen state*. Hence, if the victim nodes have similar ideal frozen states, they can mutually lock each other once they fall into these states. We give further theoretical analysis on the phenomenon in Appendix A.2. Fortunately, as shown in Figure 3(d), the ideal frozen states from different nodes are similar; therefore, it is possible to keep nodes frozen by driving their memories into similar and stable states.

**Cross-Freezing Loss.** To this end, we propose to freeze victim nodes in connected groups and make them persist frozen with mutual support from each other. We termed this approach as *Cross-Freezing*. Specifically, we first sample one-third of the victim node as the root node, then sample two of its neighbors as support nodes—note that these support neighbors also cost our attack budgets—then force their memories to minimize the loss in equation 5.

$$\mathcal{L}_u^{freeze} = \sum_{k \in N_{\text{supp}}(u)} \left( \mathcal{L}_{\text{mse}}(s_k^*, s_k^+) + \mathcal{L}_{\text{mse}}(s_u^+, s_k^+) \right) \quad (5)$$

For any given node $u$ with its memory denoted as $s_u$ and support neighbors as $N_{\text{supp}}(u)$, our objective relies on two Mean-Squared-Error (MSE) losses. The first, $\mathcal{L}_{\text{mse}}(s_k^*, s_k^+)$, aims to ensure that it updates its support neighbors' memory $s_k^+$ close to their ideal frozen state $s_k^*$ so they cannot sense future changes after the attack. We get node $k$'s ideal frozen state $s_k^*$ by repeatedly updating its memory using itself and its two support neighbors' memory until it is stabilized (i.e., has more than 0.9 cosine similarity before/after updates) or the maximum number of repeats is reached. The second loss, represented by $\mathcal{L}_{\text{mse}}(s_u^+, s_k^+)$, is designed to make sure that it updates its support neighbors' memory $s_k^+$ close to its own memory after updates (i.e., $s_u^+$) so that the messages generated between them can potentially help to lock each other and keep their memory unchanged.

### 4.2. Propagating Frozen States

**Future Simulating.** To make a node's memory influence unknown future neighbors, we propose using its existing neighbors to simulate potential future ones, which can then be used to optimize the victim nodes' ability to propagate their frozen state. This approach is based on the principle of homophily in real-world graphs, where neighboring nodes often exhibit strong similarities (McPherson et al., 2001). As an example, while applying TGN for the edge prediction tasks on Wikipedia dataset, nodes' neighbors have 0.87 cosine similarity on average, with over 60% neighboring nodes having similarities over 0.9. Hence, for a given node $u$, we simulate its future neighbors in two steps:

For a node $u$, we first augment its current neighbor set $N(u)$ by adding "presented fake future neighbors", created by sampling up to ten of the most recent neighbors and injecting Gaussian noise (mean 0, standard deviation at 0.2 times the neighbor's memory std) into their memories. Next, we simulate "newly presented fake future neighbors" to reflect brand-new nodes in the graph, initializing their memories to all zeros. The number of these new fake neighbors is proportional to the fraction of newly appeared nodes among the most recent ten neighbors of $u$. We also include more

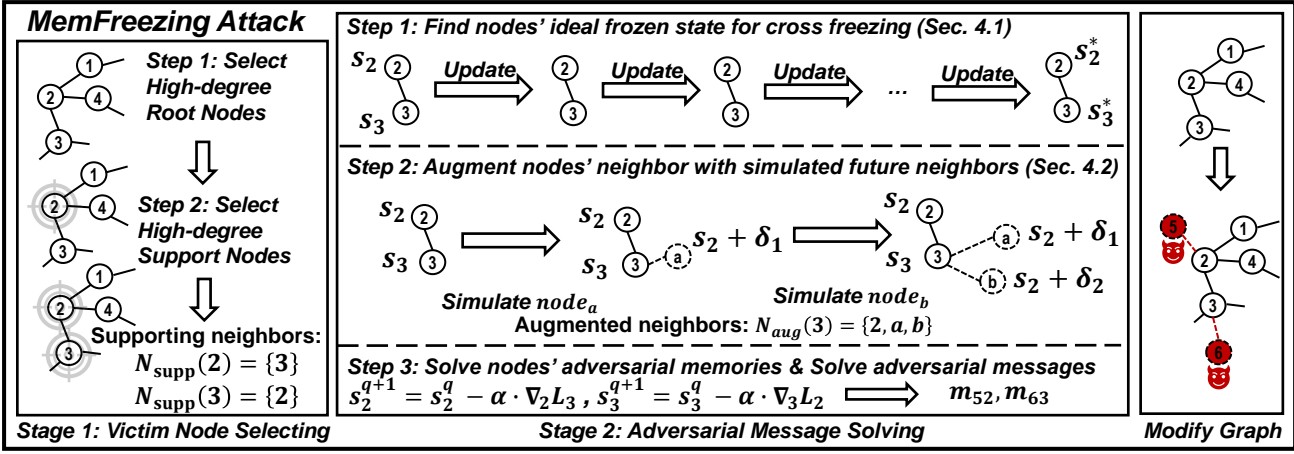

*Figure 4.* The two stages of the MemFreezing attack. In the **victim node selecting stage**, we greedily select victim nodes under the attack budget. In the **adversarial message solving stage**, we solve the victim nodes' targeted memory and corresponding adversarial messages. The solved messages are added to the graphs as fake events and removed after the attack timestamp.

details about the future simulation in Appendix A.3 and further discuss its effectiveness under extremely random and irregular graphs, where nodes have drastically changing neighbors, in Appendix B.16.

**Propagating Loss.** To make the frozen nodes contagious to potential future neighbors, we then use the resulting augmented neighbors $N_{\text{aug}}(u)$ to solve the problem described in equation 6.

$$\mathcal{L}_u^{prop} = \sum_{k \in N_{\text{aug}}(u)} \mathcal{L}_{\text{mse}}(s_u, UPDT(s_k, m_{uk})) \quad (6)$$

The objective of this loss is to minimize the Mean Squared Errors (MSEs) between a node's memory and the memories of its new neighbors after an update. By doing so, we encourage the node's memory to update its neighbors' memories (i.e., $s_k$) to become similar to itself (i.e., $s_u$).

### 4.3. Attack Framework

Combining the above-mentioned goals together, we introduce the two-stage attack framework as illustrated in Figure 4 (Detailed algorithm is presented in Appendix A.4).

**Stage 1: Victim Node Selecting.** In this stage, we use a simple greedy approach to select victim nodes in two steps: First, we select the nodes with the highest degrees in the current graph as root nodes. The intuition behind this is that we want the injected noises to be propagated to as many nodes as possible, and these high-degree nodes, such as top-commented posts on social media, are usually popular in existing and future graphs. Next, for each root node, we select its two highest-degree neighbors as its support nodes. The following procedure will treat all the root and support nodes as victim nodes and transform them into frozen states.

**Stage 2: Adversarial Message Solving.** In this stage, we

solve the adversarial event to be injected to each victim node $u$ in three steps: In the first step, we find the nodes' ideal frozen states (i.e., $s_u^*$ in equation 5) by updating its memory using current neighbors until convergence. In the second step, we simulate the future changes by augmenting victim nodes' neighbors with simulated futures (i.e., nodes/edges). The resulting neighbors are used as $N'(u)$ in equation 6. In the third step, we solve the adversarial memory $\hat{s}_u$ of the victim nodes by minimizing the total memory loss in equation 7, which is calculated by summing its persisting (i.e., equation 5) and propagating (i.e., equation 6) losses.

$$\mathcal{L}_u = \mathcal{L}_u^{freeze} + \mathcal{L}_u^{prop} \quad (7)$$

Then, we solve the adversarial messages described in equation 8 so that these messages can update the nodes' memories into their solved frozen states.

$$\arg\min_{m_{\mathcal{A}u}} \mathcal{L}_{\text{mse}}(UPDT(s_u, AGGR(m_{\mathcal{A}u}, \widetilde{m_u}), \hat{s}_u)) \quad (8)$$

The $\widetilde{m_u}$ represents the aggregated messages collected from $u$'s other neighbors. In short, for node $u$, the solution aims to find a fake message $m_{\mathcal{A}u}$ that minimizes the MSE loss between the expected noise memory $\hat{s}_u$ and the memory updated after inserting it to the graph. Lastly, for each victim node, we add the solved noisy message as a fake event from a fake node and remove it after the attack.

## 5. Evaluation

### 5.1. Experimental Setup

**Models and Datasets:** We use on four TGNN models for evaluation: JODIE (Kumar et al., 2019), Dyrep (Trivedi et al., 2019), TGN (Rossi et al., 2020) and Roland (You et al., 2022). The experiments use four dynamic graph

*Table 1.* Accumulated accuracy of edge prediction in the vanilla/attacked TGNNs over different timestamps; lower matrices indicate more effective attacks. Results on more datasets and node classification tasks are included in Appendix B.4

| Dataset | | WIKI | | | | REDDIT | | | | REDDIT-BODY | | | |
|---|---|---|---|---|---|---|---|---|---|---|---|---|---|
| Model | | TGN | JODIE | Dyrep | Roland | TGN | JODIE | Dyrep | Roland | TGN | JODIE | Dyrep | Roland |
| Vanilla | | 0.93 | 0.87 | 0.86 | 0.94 | 0.97 | 0.98 | 0.96 | 0.95 | 0.90 | 0.87 | 0.90 | 0.88 |
| $t_0$ | FN | 0.81 | 0.74 | 0.74 | 0.82 | 0.84 | 0.83 | 0.84 | 0.83 | 0.76 | 0.82 | 0.77 | 0.79 |
| | Meta-h | 0.90 | 0.83 | 0.81 | 0.85 | 0.93 | 0.95 | 0.90 | 0.92 | 0.86 | 0.83 | 0.88 | 0.85 |
| | TDGIA | **0.77** | **0.72** | **0.71** | **0.80** | **0.74** | **0.80** | **0.81** | **0.74** | **0.72** | **0.81** | **0.74** | **0.76** |
| | Ours | 0.89 | 0.78 | 0.83 | 0.87 | 0.75 | 0.84 | 0.94 | 0.82 | 0.84 | 0.85 | 0.81 | 0.78 |
| $t_{25}$ | FN | 0.92 | 0.87 | 0.85 | 0.94 | 0.97 | 0.97 | 0.96 | 0.93 | 0.90 | 0.86 | 0.89 | 0.88 |
| | Meta-h | 0.93 | 0.87 | 0.84 | 0.93 | 0.96 | 0.98 | 0.94 | 0.96 | 0.89 | 0.86 | 0.90 | 0.87 |
| | TDGIA | 0.93 | 0.81 | 0.84 | 0.92 | 0.94 | 0.95 | 0.95 | 0.90 | 0.89 | 0.85 | 0.89 | 0.88 |
| | Ours | **0.80** | **0.75** | **0.77** | **0.85** | **0.81** | **0.84** | **0.91** | **0.80** | **0.81** | **0.84** | **0.76** | **0.80** |
| $t_{50}$ | FN | 0.94 | 0.87 | 0.86 | 0.94 | 0.97 | 0.97 | 0.96 | 0.95 | 0.90 | 0.86 | 0.90 | 0.88 |
| | Meta-h | 0.93 | 0.87 | 0.85 | 0.93 | 0.97 | 0.98 | 0.94 | 0.95 | 0.90 | 0.86 | 0.90 | 0.88 |
| | TDGIA | 0.93 | 0.87 | 0.85 | 0.93 | 0.96 | 0.97 | 0.95 | 0.92 | 0.89 | 0.86 | 0.90 | 0.87 |
| | Ours | **0.75** | **0.76** | **0.75** | **0.84** | **0.80** | **0.84** | **0.91** | **0.80** | **0.77** | **0.82** | **0.76** | **0.77** |

*Figure 5.* Accumulated accuracies of TGN under no defense(left), `Adv_train`(middle), and `Lip_reg`(right) with FakeNode and our attack on `WIKI` dataset. More results in Appendix B.5

datasets: Wikipedia (`WIKI`), Reddit (`REDDIT`) (Kumar et al., 2019), Reddit-body (`REDDIT-BODY`) and Reddit-title (`REDDIT-TITLE`) (Kumar et al., 2018). More details about the models and datasets are included in Appendix B.1. We also present results on a million-node dataset, Wikipedia Talk Network (`Wiki-Talk-Temporal`) (Leskovec et al., 2010) in Appendix B.4.4.

**Tasks & Metrics:** We evaluate the models on two tasks: node classification and edge prediction (Rossi et al., 2020). For a timestamp, we measure the accuracy or area under the ROC Curve (ROC-AUC) based on all presented predictions from the beginning, which we termed as *accumulated accuracy* and *accumulated ROC-AUC*. More details about the tasks and matrices are in Appendix B.1

**Attack Setup:** We compare our work with three state-of-the-art GNN attacks: FakeNode (`FN`) (Wang et al., 2018), TDGIA(`TDGIA`) (Zou et al., 2021) and Meta-Attack-Heuristic(`Meta-h`) (Li et al., 2022). The results from Table 1 evaluate all attacks with 5% attack budgets, where we inject noises to 5% nodes of the input graph. In Appendix B.4, we evaluate attacks with 1% attack budgets. For our attack, we use a $1/3$ budget for the root nodes and $2/3$ for support nodes. All methods attack at the beginning of the test set (i.e., attack at $t_0$). We also include more details about the baseline attacks in Appendix B.2 and the results of injecting attacks in multiple timestamps in Appendix B.6.

**Defense Setup:** We adopt following defenses: Adversarial Training(`Adv_train`), Regularization under empirical Lipschitz (`Lip_reg`), and `GNNGuard` from static GNNs. More details about the defense setup are in Appendix B.3.

### 5.2. Experimental result

**Overall Performance**. We examine the accumulated accuracy at three timestamps: $t_0 = 0$, $t_{25} = 25$, and $t_{50} = 50$. The results of the edge prediction task are presented in Table 1. As observed, all prior attacks cause significant accuracy drops at $t_0$, but their impact quickly diminishes over time. By $t_{25}$ and $t_{50}$, the accumulated accuracy under these attacks is nearly identical to the baseline. In contrast, MemFreezing consistently disrupts model predictions. While it does not cause the largest accuracy drop at $t_0$ compared to other attacks due to the freezing objective, its effects are more persistent and even increase over time, achieving greater drops as the timestamps shift to $t_{25}$ and $t_{50}$. We also observe similar effects on MemFreezing when conducting the attack at different timestamps as detailed in Appendix B.7. The attacks on JODIE are less effective because JODIE employs a memory decay mechanism that uniformly decays previous memories. This introduces additional information outside the node memory, making JODIE more resilient to memory-based attacks. We discuss this phenomenon in more detail and explore potential defenses

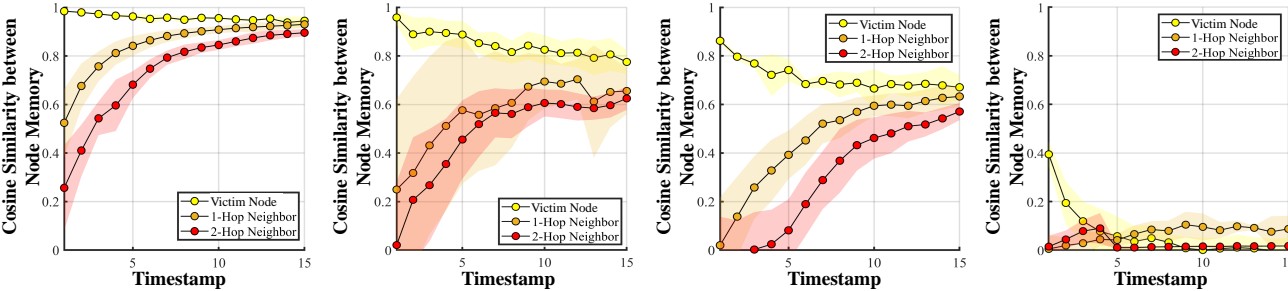

*Figure 6.* The similarities between victim nodes' initial noisy memories (at the time of the attack) and themselves'/their subsequent neighbors' memories in MemFreezing(left), MemFreezing w/o (middle-left) frozen state, MemFreezing w/o cross-freezing loss (middle-right), and regular nodes (right). All results above are from TGN and `WIKI`. More results are included in Appendix B.8.

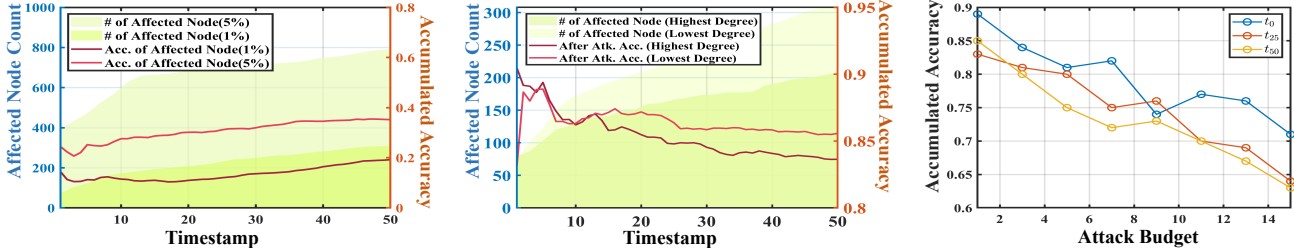

*Figure 7.* (left) Count of nodes affected by MemFreezing and accuracy for the affected nodes over time. (middle) Count of affected nodes and overall accuracy over time with two strategies for selecting the injected node: 1% lowest degree versus 1% highest degree nodes. (right) The accumulated accuracy at $t_0$, $t_{25}$, and $t_{50}$ under different attack budgets (% of total nodes). All results above are from TGN and `WIKI`. Results on more models and datasets are included in Appendix B.11.

against MemFreezing in Appendix C. We further illustrate the accumulated accuracy of TGNs under different defenses in Figure 5. Similar results are observed: the effects of baseline attacks quickly diminish, resulting in only a 1.1% accumulated accuracy drop by $t_{50}$. In contrast, MemFreezing causes progressively larger accuracy drops over time, averaging over 10% drop by $t_{50}$.

**Ablation Studies.** To analyze the propagating and persisting capability of the noise solved by MemFreezing, we capture 100 victim nodes in TGN in edge prediction on `WIKI` and monitor the changes in their memory and their neighbors' memory. In Figure 6, we compare the cosine similarity between the memories of the victim nodes at $t_0$ with those in themselves and their one-hop and two-hop neighbors at each timestamp after the attack in four versions: (1) MemFreezing, (2) MemFreezing w/o frozen state (i.e., w/o using $\mathcal{L}_{mse}(s_k^*, s_k^+)$ in equation 5), (3) MemFreezing w/o cross-freezing loss (i.e., without using entire $\mathcal{L}_u^{freeze}$ in equation 5), and (4) original TGNN without attacks. The result shows that, in MemFreezing, the noise in the victim node can persist over ten timestamps, with over 0.92 cosine similarities. For the one-hop neighbors, at $t = 1$, they achieve 0.51 average similarities after the first update by the message from victim nodes, and at $t = 15$, the average rises to 0.88. The two-hop neighbors, whose memories are updated by the message from one-hop neighbor, have average similarities that grow from 0.24 to 0.84. In contrast,

the similarity between nodes' initial attacked memory and their future counterparts drops drastically in the original TGNNs like (4). If the frozen states are not guaranteed like (2), the similarities also suffer drops and fail to achieve comparable similarities as (1). This is because, in such cases, the memories will change before reaching their converged states, making the final converged state different from the original adversarial memory states. Therefore, the converged state is essential for persisting noisy memories. The similarities drop faster if we remove the cross-freezing loss like (3) since the cross-freezing mechanism is entirely disabled. Moreover, despite removing cross-freezing losses, the neighbors are getting more similar to the target nodes, indicating that the propagating loss works as expected. We also analyzed the advances of freezing node memories compared to maximizing prediction losses in Appendix B.9 and the stealthiness of the injected noises in Appendix B.10.

**Propagation in Dynamic Graphs.** To better understand how frozen effects spread in MemFreezing, we track all topologically connected nodes to the victim node, labeling them as affected nodes since noise can potentially propagate to them. We then measure the prediction accuracy of these affected nodes (represented by colored lines). As shown in Figure 7 (left), MemFreezing progressively impacts more nodes (shaded area) and significantly reduces prediction accuracy, even though some nodes were unseen at the attack timestamp. On the one hand, attacking high-degree nodes

helps propagate the noises to more nodes, nearly doubling the number of affected nodes compared to selecting low-degree nodes, as shown in Figure 7 (middle). On the other hand, once nodes enter a stable (frozen) state, they propagate adversarial effects to future neighbors, ensuring the attack's persistence and adaptability despite dynamic graph changes.

**Scale with Attack Budget.** We also evaluate MemFreezing under broader attack budgets ranging from 1% to 15%. As shown in Figure 7 (right), higher attack budgets lead to greater accuracy drops, demonstrating MemFreezing's scalability with increased attack costs.

## 6. Conclusion

In this work, we propose MemFreezing, a novel adversarial attack tailored for TGNNs, to overcome the challenges in attacking TGNN under limited-knowledge scenarios. The MemFreezing attack misleads model predictions by freezing node memories in TGNNs into stable and dysfunctional states. The experimental results show that our approach can produce long-lasting and contagious noises in dynamic graphs, leading to significant performance drops in TGNNs.

## Acknowledgement

The authors would like to thank the anonymous ICML reviewers for their constructive feedback and suggestions. This work is supported in part by the U.S. National Science Foundation grants #2154973, #2334628, and #2312157.

## Impact Statement

Temporal Graph Neural Networks (TGNNs) have emerged as the state-of-the-art paradigm for modeling dynamic relational data in domains ranging from social networks to recommendation systems. As they become increasingly integral to real-world systems, understanding and enhancing their robustness is essential. Our work introduces MemFreezing, the first adversarial attack that assumes only limited future knowledge—a constraint inherent to live environments—and shows how subtle memory-freezing perturbations can persistently degrade TGNN performance. By exposing this practical vulnerability, we underscore the urgent need for defenses specifically designed for the dynamic, streaming nature of real-world graph applications.

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

# A. Extended Design

## A.1. Self-freezing experimental setup and theoretical analysis

We explored the viability of freezing a node by itself with a case study in TGN (Rossi et al., 2020), where the $UPDT(\cdot)$ function is typically realized using a GRU (Chung et al., 2014). At a particular timestamp, we randomly sample 100 nodes from the Wikipedia dataset and modify their memories. For each node, we use Adam optimizer (Kingma & Ba, 2014) to find a memory vector to suppress GRU updates by minimizing its reset gates (Chung et al., 2014). We then assessed if this memory state remains consistent over time.

The TGN used by the experiment uses GRU for memory updating (i.e., for implementing $UPDT(\cdot)$ function in equation 3), as depicted in equation 9-12.

$$r_t = \sigma(W_{ir}\widetilde{m_t} + b_{ir} + W_{hr}s_{t-1} + b_{hr}) \tag{9}$$
$$z_t = \sigma(W_{iz}\widetilde{m_t} + b_{iz} + W_{hz}s_{t-1} + b_{hz}) \tag{10}$$
$$n_t = tanh(W_{in}\widetilde{m_t} + b_{in} + r_t \odot \sigma(W_{in}\widetilde{m_t} + b_{in})) \tag{11}$$
$$s_t = (1 - z_t) \odot n_t + z_t \odot s_{t-1} \tag{12}$$

where $\sigma(\cdot)$ is the sigmoid function. Given the node memory $s_{t-1} \in \mathbb{R}^M$ at the previous timestamp, and the aggregated message $\widetilde{m_t} \in \mathbb{R}^D$ at time $t$, GRUs first compute reset gate $r_t \in \mathbb{R}^M$, update gate $z_t \in \mathbb{R}^M$, and new gate $n_t \in \mathbb{R}^M$.

In this experiment, we aim to minimize the interference of the message, $\widetilde{m_t}$, and maintain the updated memory, $s_t$, close to the previous memory, $s_{t-1}$. To this scope, we can maximize all the features in the update gate, $z_t$, until it approaches $\mathbb{1}$, where the update gate will be directly used to control the portion of the previous memory, which is:

$$\text{as } z_t \to \mathbb{1}, \quad s_t \to \mathbb{0} \odot n_t + \mathbb{1} \odot s_{t-1} \approx s_{t-1} \tag{13}$$

Additionally, according to Equation 10, the update gate $z_t$ is computed by the sum of two linear processes, and one is from the message, $\widetilde{m_t}$ and the other one is from memory $s_{t-1}$. As we maximize the linear output of the memory, $W_{hz} \cdot s_{t-1}$, the update gate, $z_t$, is then maximized.

Hence, to analyze the maximum output of the linear process, $W_{hz} \cdot s_{t-1}$, we formulate it into a linear program problem with the equations:

$$\max \sum W_{hz} \cdot s_{t-1}$$
$$\text{s.t.} \ -\mathbb{1} \le s_{t-1} \le \mathbb{1} \tag{14}$$
$$W_{hz} \cdot s_{t-1} > \delta$$

As the memory is the output of the $tanh$ function rather than the unit-length vector, $s_t$ is bounded by the limit of the $tanh$ function, $[-1, 1]^M$. Further, we introduce an addition constraint $W_{hz} \cdot s_{t-1} > \delta$ to guarantee all dimensions of the linear output are bound by a constant, $\delta$.

The optimal result for the memory, $s_{t-1}^*$, for the linear problem only depends on the model weights, where given a TGN model, the solution of the self-freezing memory is unique, and we have conducted the experiment on three models TGN+WIKI, TGN+REDDIT, and a randomly initialized model.

The result in Figure 8-10 (a) shows the maximum update gate, $z_t^*$, computed by $\sigma(W_{hz} \cdot s_{t-1}^*)$. In the TGN+wiki example, $z_t^*$ is a 172-dimension vector, and it is distributed with a mean of 0.64 and a standard deviation of 0.12. As aforementioned, to achieve the self-freezing memory, the update gate, $z_t$, is required to approach $\mathbb{1}$, but it is infeasible to fine the solution in the real world case under the constraints. In Figure 8-10 (b), we simulate the GRU updating starting with the optimal memory, $s_{t-1}^*$, and monitor the cosine similarity between the memory before updated and after updated. The results further demonstrate even the optimal solution cannot accomplish the self-freezing goal.

To theoretically analyze the maximum of the in the general case, we divide them into their eigen-representations, and we use the SVD decomposition:

$$W_{hz} = U \cdot \Sigma \cdot V^T = \sum_i e_i \cdot U_i \cdot V_i^T, \quad s_{t-1} = \sum_{i|V_i \in V} \alpha_i \cdot V_i \tag{15}$$

In SVD decomposition, $U$ and $V$ are the unitary matrix, and we use the basis from $V$ to decompose $s_{t-1}$. Moreover, the linear process is written as:

$$W_{hz} \cdot s_{t-1} = \sum_i e_i \cdot \alpha_i \cdot U_i \cdot V_i^T \cdot V_i = \sum_i \alpha_i \cdot e_i \cdot U_i \qquad (16)$$

This linear process is represented by the linear combination on the basis of $U$. We can easily acquire the theoretical maximum of the output. As $s_{t-1} \in [-1, 1]^M$, if $V_\theta$ is a basis of $\{-\frac{1}{\sqrt{M}}, \frac{1}{\sqrt{M}}\}^M$, $s_{t-1} = s_\theta$ can achieve the maximum projection to this basis, which is, $\alpha_\theta = \sqrt{M}$ and $s_{t-1} = V_\theta$. Similarly, the linear output $W_{hz} \cdot s_{t-1}$ achieves the maximum by there exist a basis $U_\theta = \{\frac{1}{\sqrt{M}}\}^M$, and the linear output,

$$W_\theta \cdot s_\theta = e_\theta \cdot \frac{1}{\sqrt{M}} \cdot \sqrt{M} \cdot \mathbb{1} = e_\theta \cdot \mathbb{1} \qquad (17)$$

As is shown, the maximum output of the linear process is equal to the eigenvalue. According to the experiment, the largest eigenvalue of the weight matrix is usually around 2. Therefore, the update gate, $z_t$, has the theoretical maximum value, $\sigma(e_\theta) \approx 0.88$.

However, the weights, $W_{hz}$, are trained through the model update, which makes it impossible to find the ideal maximum in the practical case.

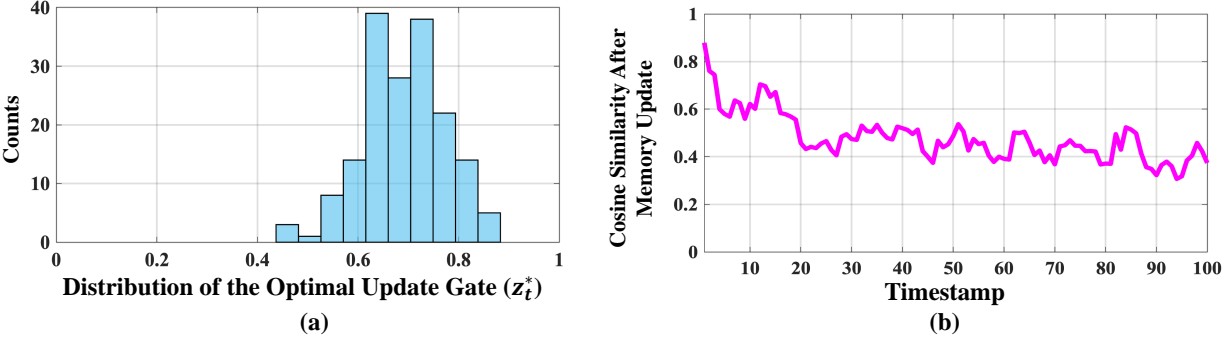

(a)            (b)

*Figure 8.* (a) The distribution of the optimal update gate $z_t^*$. (b) The cosine similarity between memory before the update and after the update, starting with the optimal self-freezing memory $s_t^*$. Experiments are conducted in the TGN model with `WIKI` datasets.

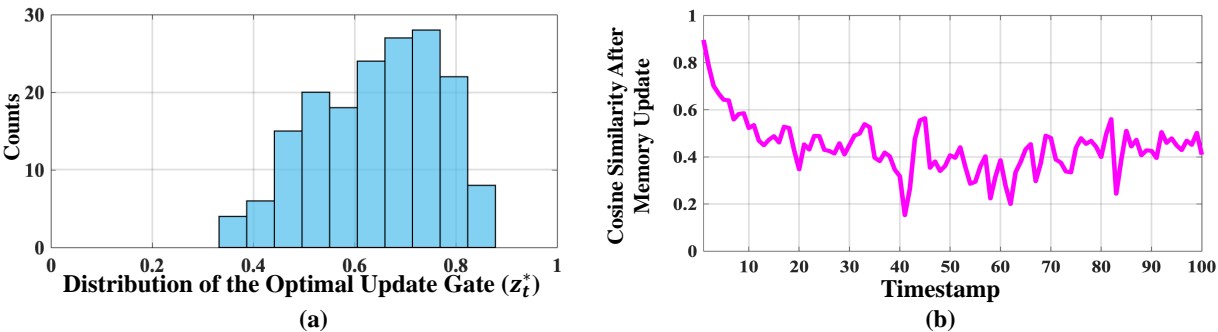

(a)            (b)

*Figure 9.* (a) The distribution of the optimal update gate $z_t^*$. (b) The cosine similarity between memory before the update and after the update, starting with the optimal self-freezing memory $s_t^*$. Experiments are conducted in the TGN model with `REDDIT` datasets.

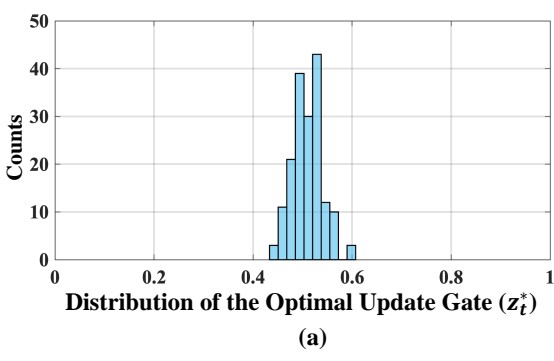
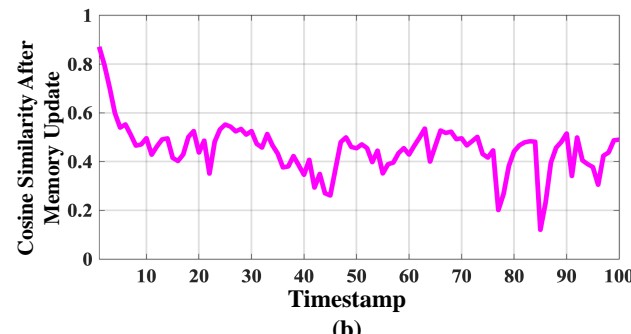

*Figure 10.* (a) The distribution of the optimal update gate $z_t^*$. (b) The cosine similarity between memory before the update and after the update, starting with the optimal self-freezing memory $s_t^*$. Experiments are conducted in the randomly sampled GRU model.

### A.2. Theoretical analysis for memory under cross-freezing

In Appendix A.1, we show that the self-freezing will be hard to achieve as the update gate $z_t$ is hard to achieve all ones, $\mathbb{1}$. The update gate is computed by, $z_t = \sigma(W_{iz}\widetilde{m_t} + b_{iz} + W_{hz}s_{t-1} + b_{hz})$, and in equation (14), we only consider the self-freeze term, $W_{hz}s_{t-1}$. For the cross-freezing case, we introduce two or more supporting nodes, and thus, we can jointly maximize the first two terms for the update gate, $W_{iz}\widetilde{m_t} + W_{hz}s_{t-1}$, and the objective function can be written as:

$$\max(W_{iz1} + W_{iz2})(\epsilon \odot s) + W_{hz}s$$
$$s.t. \quad \mathbb{1} \le s \le \mathbb{1} \tag{18}$$

The message has four variable concatenated: $m_t = [s_{\text{src}}, s_{\text{dst}}, e, t]$, and the matrix product, $M_{iz}m = W_{iz1}s_{\text{src}} + W_{iz2}s_{\text{dst}} + W_{iz3}e + W_{iz3}t$. In this object function, we assume in the cross-freeze state, the source and destination memory converge to a certain optimal, $s^*$. Consider that there might be other non-supported nodes contributing to the message, $\widetilde{m}$, so we model this by a coefficient $\epsilon$ vector, where $\epsilon \odot s = \widetilde{s}$. As the support node become the majority among the connected nodes, the coefficient vector $\epsilon \to 1$.

To simplify this problem, we rewrite the equation as:

$$(W_{iz1} + W_{iz2})(E[\epsilon] \cdot s) + W_{hz}s + \mathcal{O}(1)$$
$$= (E[\epsilon] \cdot W_{iz1} + E[\epsilon] \cdot W_{iz2} + W_{hz})s + \mathcal{O}(1) \tag{19}$$

The above equation will be similar to Equation (14), as considering $(\mathbb{E}[\epsilon] \cdot W_{iz1} + \mathbb{E}[\epsilon] \cdot W_{iz2} + W_{hz})$ as a new matrix. Let the weights in the GRU unit follows the Gassuian distribution, $w \in N(0, \frac{1}{\sqrt{M}})$, then the new matrix would follow, $w' \in N(0, \frac{\sqrt{2\mathbb{E}[\epsilon]^2+1}}{\sqrt{M}})$. Hence, the optimal basis will then be, $\{-\frac{\sqrt{2\mathbb{E}[\epsilon]^2+1}}{\sqrt{M}}, \frac{\sqrt{2\mathbb{E}[\epsilon]^2+1}}{\sqrt{M}}\}^M$, where its egienvalue can achieve $\sqrt{2\mathbb{E}[\epsilon]^2 + 1}$ times larger than self-freeze one.

As a result, the output update gate, $z_t$, will scale accroding to $\mathbb{E}[\epsilon]$, and here is the list of the outcome:

$$\mathbb{E}[\epsilon] = 0.5, \ z_t^* = 0.919$$
$$\mathbb{E}[\epsilon] = 0.75, \ z_t^* = 0.94 \tag{20}$$
$$\mathbb{E}[\epsilon] = 1, \ z_t^* = 0.96$$

Hence, it will become easier to freeze the memory by introducing the support nodes

### A.3. Details of simulating fake future neighbors

To simulate the potential future neighbors of the victim nodes and enhance their capability to contaminate those nodes, we randomly sample existing neighbors from victim nodes' and add Gaussian noise to their features. For the mean of the Gaussian noise, we use 0 as the mean for all nodes. We use 0.2 times the standard variation of the original neighbor's memory for the standard variations of the Gaussian noise. In summary, for a node $v$, we follow Equation equation 21 to simulate a victim node's neighbors,

$$s_i' = s_i + \mathcal{N}(0, \eta \cdot \sigma(s(v))), i \in N(v) \tag{21}$$

In which $s'_i$ stands for the fake future neighbors and $s_i$ stands for the memories from a sampled existing neighbor, $N(v)$ indicates the current neighbor set of node $v$. For the Gaussian noise $\mathcal{N}(0, \eta \cdot \sigma(s(v)))$, it has meant as 0, $\eta = 0.2$, and $\sigma(s(v))$ as the standard variation of all features in existing neighbors.

We use the $\Delta T$ of the most recent clean message on the victim nodes for the timestamp of their appearances. For example, if we attack node n, whose most recent message before our attack uses $\Delta T_k$ at its updating, then the timestamp of the fake future neighbors will also be $\Delta T_k$. It is also worth mentioning that, the $\Delta T$ has limited effects on the updating process. As proposed in TGAT and used in TGN and other TGNN models, the $\Delta T$ is encoded into a time vector first as

$$E(\Delta T) = W * cos(\Delta T)$$

in which $W$ is a weight vector with 172 dimensions with descending magnitudes (for example, $[1.00, 0.88, 0.78, ...1.12e - 09, 9.99e - 10]$). Then, $W$ is used to update the memory. The value of $W$ is very small except for the first few dimensions, making them can hardly affect the updating process.

## A.4. Overall Algorithm

---

**Algorithm 1 MemFrezzing Attack**

---

**Input** : $\mathcal{G} = (V, s(V)) \leftarrow$ Original graph with Node $V$, memories $s(V)$
**Input** : $\forall_{i,j|V_i,V_j \in V} \quad m(s_i, s_j, e_{ij}, \Delta t) \leftarrow$ Messages before $t_0$
**Input** : $\mathcal{B} \leftarrow$ Number of attacked nodes (attack budget)
**Input** : $q \leftarrow$ Number of support neighbors for each root node.
**Input** : $N(V), N_{\text{supp}}(V), N_{\text{aug}}(V) \leftarrow$ the full neighbors sets, supported neighbors, and augmented neighbors set
**Output** : $V_{\mathcal{A}}, e_{\mathcal{A}}$: Perturbed nodes and message.

/* **Stage 1. Victim Node Sampling** */
$n \leftarrow \mathcal{B}/(q+1)$
$V_1^{\text{root}}, V_2^{\text{root}}, \cdots, V_n^{\text{root}} \leftarrow topk(\text{degree}(V), n)$
**for** $i \in \{1, 2, \cdots, n\}$ **do**
 $\mathbf{V}^{\text{support}} \leftarrow V_i^{\text{root}} \cup \{V_1, V_2, \cdots V_q \in N(V_i^{root})\}$
 $\{\hat{s}_0, \hat{s}_1, \cdots \hat{s}_q\} \leftarrow \texttt{ComputeConvergeState}(\mathbf{V}^{support})$
 $\mathbf{V}_{\mathcal{A}}, \mathbf{e}_{\mathcal{A}} \leftarrow \texttt{ComputeAdversarialMessage}(\{\hat{s}_0, \hat{s}_1, \cdots \hat{s}_q\}, \mathbf{V}^{support})$

/* **Stage 2. Solving Frozen State** */
**Function** $\texttt{ComputeConvergeState}(\mathbf{V}^{support})$

 /* **2.1. Solving the Ideal Frozen State** */
 **for** $i \mid V_i \in \mathbf{V}^{support}$ **do**
  $s_i \leftarrow s(V_i)$
  $m \leftarrow m(s_i, s_i, e_{ij}, \Delta t)$ , where $j \mid V_j \in \mathbf{V}^{\text{support}}$
  **do**
   $s_i \leftarrow s_i^+$
   $m \leftarrow m(s_i, s_i, e_{ij}, \Delta t)$
   $s_i^+ \leftarrow UPDT(s_i, m)$
  **while** $\|s_i^+ - s_i\|_2^2 > \epsilon$;
 $s_1^*, s_2^*, \cdots s_q^* \leftarrow s_1^+, s_2^+, \cdots s_q^+$

 /* **2.2. Solving the Cross-Frozen State** */
 $s_1^{(0)}, s_1^{(0)}, \cdots s_q^{(0)} \leftarrow s_1^*, s_2^*, \cdots s_q^* + \mathcal{N}(0, \eta \cdot \sigma(s(V)))$
 **for** $t \in \{0, 1, 2, \cdots, T\}$ **do**
  $\forall_{i \in \{1,2,\cdots q\}} , \quad s_i^{(t)+} \leftarrow UPDT(s_i^{(t)}, \widetilde{m_i})$
  **for** $i \in \{0, 1, 2, \cdots, q\}$ **do**
   $\mathcal{L}_i^{freeze} \leftarrow \sum_{k \in N_{\text{supp}}(i)} \left( \mathcal{L}_{\text{mse}}(s_k^{(t)+}, s_k^*) + \mathcal{L}_{\text{mse}}(s_i^{(t)+}, s_k^{(t)+}) \right)$
   $\mathcal{L}_i^{prop} \leftarrow \sum_{k \in N_{\text{aug}}(i)} \mathcal{L}_{\text{mse}}(s_i^{(t)}, UPDT(s_i^{(t)}, m_{ik}))$
  $\forall_{i \in \{1,2,\cdots q\}} , \quad s_i^{(t+1)} \leftarrow s_i^{(t)+} - \alpha \cdot \nabla_{s_i}(\mathcal{L}_i^{freeze} + \mathcal{L}_i^{prop})$
 **return** $\{s_0^{(T)}, s_1^{(T)}, s_2^{(T)}, \cdots s_q^{(T)}\}$

/* **Stage 3. Solving the Adversarial Message** */
**Function** $\texttt{ComputeAdversarialMessage}(\{\hat{s}_0, \hat{s}_1, \cdots \hat{s}_q\}, \mathbf{V}^{support})$

 **for** $V_i \in \mathbf{V}^{support}$ **do**
  $V_{i,\mathcal{A}} \leftarrow V \mid V \in N(V_i)$
  **for** $t \in \{0, 1, \cdots, T\}$ **do**
   $m_{\mathcal{A}i}^{(t)} \leftarrow m(s_i, s(V_{\mathcal{A}}), e_{\mathcal{A}i}^{(t)}, \Delta t)$
   $\mathcal{L}_{\mathcal{A}} \leftarrow \mathcal{L}_{\text{mse}}(UPDT(s_i, AGGR(m_{\mathcal{A}i}^{(t)}, \widetilde{m_i}), \hat{s}_i)$
   $e_{\mathcal{A}i}^{(t+1)} \leftarrow e_{\mathcal{A}i}^{(t)} - \alpha \cdot \nabla_{e_{\mathcal{A}i}^{(t)}} \mathcal{L}_{\mathcal{A}}$

 **return** $\{V_{1,\mathcal{A}}, V_{2,\mathcal{A}}, \cdots, V_{q,\mathcal{A}}\} , \{e_{\mathcal{A}1}, e_{\mathcal{A}2}, \cdots, e_{\mathcal{A}q}\}$

---

# B. Extended Evaluation

## B.1. Experimental Details.

**Model Details**. All four TGNNs we included maintain a memory vector in each node and follow the memory updating process as discussed in Section 2. And they are different in their node embedding procedure (i.e., equation 4). Specifically, Dyrep directly uses the node memories for the predictions (i.e., $h_i^t = s_i^t$). JODIE applies a time-decay coefficient to the scale memories before classification (i.e., $h_i^t = \delta(t) \cdot s_i^t$). TGN, on the other hand, refines memories using a single-layer graph attention module, as outlined in equation 4. Unlike prior models, ROLAND (You et al., 2022) is a recent model designed for DTDG graphs, yet it also maintains a history node feature for each node as memory. Specifically, it adopts a multi-layer memory mechanism by keeping memory for both memory and embedding stages. In other words, for the graph embedding part, it also adopts a GRU to combine nodes' previous embedding with the current embedding gathered from updated node memories. All the models update and embed memory for one time at each prediction (i.e., one layer aggregation in equation 3 and equation 4). The node memory dimension is set to 172, and the node embedding dimension is set to 100. Following the training steps in (Rossi et al., 2020), we use Adam optimizer with learning rate $\alpha = 0.01$ to train the models 120 epochs.

**Tasks Details.** Models for node classification are trained to predict binary labels on each node. We use the commonly used Area under the ROC Curve (ROC-AUC) to measure the model performances. The models for edge prediction are self-supervise trained, using the edge information in future steps. During the testing, given a source node, they predict the possibility of whether another node will be its next incoming destination node and then decide which node will be its next neighbor. We use prediction accuracy for evaluating the edge prediction result.

**Dataset Details.** Reddit and Wikipedia are dynamic interaction graphs retrieved from online resources in (Rossi et al., 2020). In Wikipedia datasets, the nodes represent users and wiki pages, and the edges indicate editing from users to pages. In the Reddit dataset, the nodes represent users and subreddits, and an edge within it represents a poster from a user posted on a subreddit. The edge features are represented by text features, and the node labels indicate whether a user is banned. All the abovementioned information is accompanied by timestamps. Align with their original designs (Kumar et al., 2019), and we set the newly input nodes' features as zero feature vectors. Reddit-body and Reddit-title are two larger-scale datasets that represent the directed connections between two subreddits (a subreddit is a community on Reddit). The dataset is collected by SNAP using publicly available Reddit data of 2.5 years from Jan 2014 to April 2017 (Kumar et al., 2018). The statistics of the dataset used are shown in Table 2.

*Table 2.* Dataset details

|  | # of Nodes | # of Edges | # Edge Feature | # of Node Feature |
|---|---|---|---|---|
| **Wikipedia(`WIKI`)** | 9,227 | 157,474 | 172 | 172 |
| **Reddit(`REDDIT`)** | 11,000 | 672,447 | 172 | 172 |
| **Reddit-Body(`REDDIT-BODY`)** | 35,776 | 286,561 | 64 | 172 |
| **Reddit-Title(`REDDIT-TITLE`)** | 54,075 | 571,927 | 64 | 172 |

**Platform details.** We list then environment details in Table 3.

*Table 3.* Experimental Environment Setting

| Environment | Details |
|---|---|
| **OS** | Windows 11 |
| **CPU** | Intel i9-13900K |
| **Memory** | 64GB DDR5 RAM |
| **GPU** | NVIDIA RTX 4090 |
| **Platform** | PyTorch 2.2.1 |
| **CUDA Version** | CUDA 12.1 |

## B.2. Baseline attack and attack setup

We adopt the following attacks toward static GNNs. Specifically, we adopt the attack at the same time as our attack time by attacking the existing dynamic graph as a static graph:

**FakeNode (Wang et al., 2018)** uses a greedy approach to generate edges of malicious nodes and their corresponding features

to mislead the static GNN predictions. Note that this approach assumes that the added nodes/edges will be kept in the graph, so we keep the fake nodes and edges still after the attack timestamp. Differently, the attacking nodes in MemFrezzing are removed after the attack.

**TDGIA (Zou et al., 2021)** is a cutting-edge Graph Injection Attack tailored to compromise static GNNs. This method exploits the inherent vulnerabilities of GNNs and the unique topological characteristics of graphs. In our implementation for each target node, we adhere to the established methodology of TDGIA to identify the top 65% susceptible edges, utilizing their specialized scheme for selecting topologically defective edges. These edges are then optimized using gradient descent. Notably, the scale of modifications applied to each target node in the TDGIA method is substantially larger than our approach, involving adjustments to 65% edges per node instead of just one edge per node. Furthermore, these modifications will be kept after the attack instead of being removed as our attack.

**Meta_Attack_Heuristic (Li et al., 2022)** is a heuristic-based attack inspired by the meta attack (Zügner & Günnemann, 2019). This heuristic-based approach is an evolution of the original meta-attack, which relied on gradient-based edge selection. The updated heuristic version demonstrates greater versatility across a variety of GNN models and large-scale graphs, and it exhibits enhanced effectiveness compared to its predecessor. Notably, the meta-attack and its heuristic counterpart operate under the assumption that edges lack attributes. Consequently, in our application, we assign an all-zero feature to the fake edges inserted as part of the attack process.

For all attacks (including our attack), We select ranges of noisy messages (i.e., magnitudes of message features ) between -1 and 1 since -1 and 1 are the theoretical minimum and maximum values of the clean messages. The messages in TGNNs are usually memories of the nodes updated from previous timestamps, which have activation functions such as tanh/cosine functions right before the outputs. Therefore, all features of these messages (i.e., memories) should be within the range of -1 and 1 as the minimum and maximum values of the activation functions (i.e., tanh). Therefore, using -1 and 1 produces messages that are exactly similar to those of the other features in the graph.

All adversarial messages/nodes in the baselines and our attacks use the $\Delta T$ of the most recent clean message on the victim nodes. For example, if we attack node n, whose most recent message before our attack uses $\Delta T_k$ at its updating, then the timestamp of the fake messages added to this node will be $\Delta T_k$ as well. It is also worth mentioning that the delta T has limited effects on the updating process, as we discussed in Appendix A.3.

For all attacks, we define the attack budget as the ratio of nodes that are affected. To ensure a fair comparison, all attacks target the same set of victim nodes (the highest-degree ones). We would also like to mention that, although targeting these high-degree nodes, all benchmarked attacks, including MemFreezing, either inject one-degree nodes or edges into the graph and affect the same number of victim nodes at the time of the attack.

Specifically, MemFreezing targets high-degree nodes by introducing a temporary fake node for each target and creating an event (i.e., an edge) between the fake node and the target. In this way, MemFreezing, like FakeNode, injects nodes with a degree of one into the graph. However, unlike FakeNode, which retains the injected fake nodes and can potentially cause stronger adversarial effects, MemFreezing removes these fake nodes after the attack, minimizing structural changes while inducing long-lasting adversarial effects. Therefore, given a graph with $V$ nodes and $E$ edges and targeting $N = 5\%V$ victim nodes (i.e., 5% budget), MemFreezing adds $N$ fake edges. Since nodes typically have a degree greater than one, $K = 5\%E > 5\%V = N$, the edge changes are less than 5% edges.

## B.3. Baseline defenses setup

We adopt the following defensive strategies for the vanilla TGNN models:

**Adversarial Training:** In line with the approach detailed in (Madry et al., 2017), we introduce perturbations to the node memories in TGNN models during the training. We then employ a minimax adversarial training scheme to enhance the robustness of the TGNN model against these perturbations.

**Regularization under empirical Lipschitz bound:** Following the methodology in (Jia et al., 2023), we minimize the empirical Lipschitz bound during the TGNN training process, where the empirical Lipschitz bound, $L$, is computed by:

$$L = \sup_{\Delta} \ \frac{||f(x + \Delta) - f(x)||_2^2}{||\Delta||_2^2} \tag{22}$$

This regularization aims to bound the effectiveness of small perturbations, such as adversarial examples.

**GNNGurad:** Following the insights that only the similar node may provide significant information for prediction, GNN-Guard(Zhang & Zitnik, 2020) adopts a cosine-similarity-based approach to discount the messages passing between dissimilar nodes.

Notably, most robust GCN models, such as RobustGCN, SGCN, GraphSAGE, and TAGCN mentioned in (Zou et al., 2021), are primarily tailored for static graph benchmarks. Given their design constraints, these models are unsuited for TGNN setup with dynamic graph benchmarks and do not offer a viable defense for the TGNN models targeted by our attack.

## B.4. Extra Main Results

Here we report edge prediction accuracies on REDDIT-TITLE in Table 4, and node classification AUCs on WIKI in Table 5. The results indicate that: (1) The static attacks cannot last long and affect future nods. (2) Our approach can be more and more effective after the attack time.

### B.4.1. EDGE PREDICTION RESULTS (5% ATTACK BUDGET)

Here, we report edge prediction accuracies on REDDIT-TITLE in Table 4. The results indicate that: (1) The static attacks cannot last long and affect future nods. (2) Our approach can be more and more effective after the attack time.

Table 4. Accumulated accuracy of edge prediction in the vanilla/attacked TGNNs over different timestamps on REDDIT-TITLE; lower matrices indicate more effective attacks.

| Dataset | | REDDIT-TITLE | | | |
|---------|--------|------|-------|-------|--------|
| Model | | TGN | JODIE | Dyrep | ROLAND |
| Vanilla | | 0.93 | 0.92 | 0.91 | 0.91 |
| $t_0$ | FN | 0.76 | 0.82 | 0.77 | 0.79 |
| | Meta_h | 0.86 | 0.83 | 0.88 | 0.85 |
| | TDGIA | **0.72** | **0.81** | **0.74** | **0.76** |
| | ours | 0.84 | 0.85 | 0.81 | 0.78 |
| $t_{25}$ | FN | 0.9 | 0.86 | 0.89 | 0.88 |
| | Meta_h | 0.89 | 0.86 | 0.9 | 0.87 |
| | TDGIA | 0.89 | 0.85 | 0.89 | 0.88 |
| | ours | **0.81** | **0.84** | **0.76** | **0.80** |
| $t_{50}$ | FN | 0.9 | 0.86 | 0.9 | 0.88 |
| | Meta_h | 0.9 | 0.86 | 0.9 | 0.88 |
| | TDGIA | 0.89 | 0.86 | 0.9 | 0.87 |
| | ours | **0.77** | **0.82** | **0.76** | **0.77** |

### B.4.2. NODE CLASSIFICATION RESULTS (5% ATTACK BUDGET)

Here, we report node classification AUCs on WIKI in Table 5. The results are similar to the edge predictions: Static attacks are good at the first attack time but cannot last long and affect future nods. In contrast, MemFrezzing can be more and more effective after the attack time.

Table 5. The AUC of vanilla/attacked TGNNs on the node classification task; lower matrices indicate more effective attacks.

| Dataset | | WIKI | | | |
|---------|-------|------|-------|-------|--------|
| Model | | TGN | JODIE | Dyrep | ROLAND |
| Vanilla | | 0.90 | 0.88 | 0.89 | 0.90 |
| t0 | FN | **0.77** | 0.87 | **0.75** | 0.78 |
| | Meta | 0.86 | 0.83 | 0.86 | 0.85 |
| | TDGIA | 0.73 | **0.82** | 0.76 | **0.75** |
| | ours | 0.82 | 0.88 | 0.84 | 0.80 |
| t25 | FN | 0.90 | 0.88 | 0.88 | 0.88 |
| | Meta | 0.89 | 0.87 | 0.88 | 0.89 |
| | TDGIA | 0.88 | 0.87 | 0.88 | 0.89 |
| | ours | **0.82** | **0.85** | **0.81** | **0.79** |
| t50 | FN | 0.90 | 0.88 | 0.88 | 0.90 |
| | Meta | 0.90 | 0.89 | 0.90 | 0.90 |
| | TDGIA | 0.90 | 0.88 | 0.88 | 0.90 |
| | ours | **0.80** | **0.85** | **0.77** | **0.77** |

### B.4.3. RESULTS WITH DIFFERENT BUDGET(1% ATTACK BUDGET)

To more comprehensively show the impact of the attack budget, we include detailed results of baselines' and our attacks' effectiveness under the attack budget as 1%. As shown in Table 6, Table 7, and Table 8, our approach can outperform baselines as well, despite fewer nodes being attacked.

*Table 6.* Accumulated accuracy of edge prediction in the vanilla/attacked TGNNs over different timestamps on WIKI and REDDIT; The attack budget is 1% for all attacks; lower matrices indicate more effective attacks.

| Dataset | | WIKI | | | | REDDIT | | | |
|---------|-------|------|-------|-------|--------|------|-------|-------|--------|
| Model | | TGN | JODIE | Dyrep | ROLAND | TGN | JODIE | Dyrep | ROLAND |
| Vanilla | | 0.93 | 0.87 | 0.86 | 0.94 | 0.97 | 0.98 | 0.96 | 0.95 |
| $t_0$ | FN | 0.89 | 0.83 | 0.82 | 0.85 | 0.93 | 0.93 | 0.92 | 0.85 |
| | Meta | 0.92 | 0.85 | 0.83 | 0.89 | 0.95 | 0.96 | 0.94 | 0.93 |
| | TDGIA | 0.83 | 0.81 | 0.77 | 0.83 | 0.89 | 0.88 | 0.88 | 0.8 |
| | ours | 0.9 | 0.82 | 0.84 | 0.9 | 0.93 | 0.94 | 0.94 | 0.86 |
| $t_{25}$ | FN | 0.92 | 0.87 | 0.85 | 0.94 | 0.97 | 0.97 | 0.96 | 0.95 |
| | Meta | 0.93 | 0.86 | 0.85 | 0.93 | 0.95 | 0.98 | 0.95 | 0.94 |
| | TDGIA | 0.91 | 0.84 | 0.83 | 0.93 | 0.94 | 0.96 | 0.96 | 0.92 |
| | ours | 0.8 | 0.82 | 0.82 | 0.88 | 0.81 | 0.84 | 0.91 | 0.84 |
| $t_{50}$ | FN | 0.94 | 0.87 | 0.86 | 0.94 | 0.97 | 0.97 | 0.96 | 0.95 |
| | Meta | 0.94 | 0.87 | 0.86 | 0.93 | 0.96 | 0.98 | 0.95 | 0.95 |
| | TDGIA | 0.94 | 0.87 | 0.85 | 0.93 | 0.96 | 0.97 | 0.95 | 0.93 |
| | ours | 0.85 | 0.81 | 0.80 | 0.86 | 0.83 | 0.84 | 0.91 | 0.83 |

*Table 7.* Accumulated accuracy of edge prediction in the vanilla/attacked TGNNs over different timestamps on REDDIT-BODY and REDDIT-TITLE; The attack budget is 1% for all attacks; lower matrices indicate more effective attacks.

| Dataset | | REDDIT-BODY | | | | REDDIT-TITLE | | | |
|---------|-------|-------------|-------|-------|--------|--------------|-------|-------|--------|
| Model | | TGN | JODIE | Dyrep | ROLAND | TGN | JODIE | Dyrep | ROLAND |
| Vanilla | | 0.9 | 0.87 | 0.9 | 0.88 | 0.93 | 0.92 | 0.91 | 0.91 |
| $t_0$ | FN | 0.85 | 0.85 | 0.81 | 0.83 | 0.88 | 0.88 | 0.85 | 0.83 |
| | Meta | 0.87 | 0.85 | 0.87 | 0.86 | 0.92 | 0.89 | 0.89 | 0.9 |
| | TDGIA | 0.81 | 0.83 | 0.79 | 0.78 | 0.85 | 0.87 | 0.85 | 0.83 |
| | ours | 0.87 | 0.85 | 0.85 | 0.82 | 0.88 | 0.9 | 0.86 | 0.85 |
| $t_{25}$ | FN | 0.9 | 0.84 | 0.89 | 0.88 | 0.92 | 0.92 | 0.9 | 0.91 |
| | Meta | 0.9 | 0.87 | 0.9 | 0.88 | 0.93 | 0.93 | 0.91 | 0.91 |
| | TDGIA | 0.88 | 0.86 | 0.9 | 0.87 | 0.92 | 0.92 | 0.9 | 0.91 |
| | ours | 0.84 | 0.86 | 0.8 | 0.82 | 0.85 | 0.88 | 0.81 | 0.86 |
| $t_{50}$ | FN | 0.9 | 0.87 | 0.9 | 0.88 | 0.93 | 0.92 | 0.9 | 0.91 |
| | Meta | 0.9 | 0.88 | 0.9 | 0.88 | 0.93 | 0.93 | 0.9 | 0.91 |
| | TDGIA | 0.89 | 0.87 | 0.9 | 0.87 | 0.93 | 0.91 | 0.9 | 0.9 |
| | ours | 0.79 | 0.85 | 0.77 | 0.83 | 0.8 | 0.83 | 0.82 | 0.83 |

*Table 8.* The AUC of vanilla/attacked TGNNs on the node classification task under 1% node attacked budget; lower matrices indicate more effective attacks.

| Dataset | | TGN | WIKI JODIE | Dyrep | ROLAND |
|---|---|---|---|---|---|
| | Model | TGN | JODIE | Dyrep | ROLAND |
| | Vanilla | 0.9 | 0.88 | 0.89 | 0.9 |
| $t_0$ | FN | 0.83 | 0.88 | 0.83 | 0.83 |
| | Meta | 0.87 | 0.85 | 0.88 | 0.88 |
| | TDGIA | 0.81 | 0.85 | 0.83 | 0.8 |
| | ours | 0.86 | 0.88 | 0.86 | 0.85 |
| $t_{25}$ | FN | 0.89 | 0.88 | 0.89 | 0.9 |
| | Meta | 0.9 | 0.88 | 0.88 | 0.89 |
| | TDGIA | 0.9 | 0.87 | 0.89 | 0.89 |
| | ours | 0.82 | 0.85 | 0.81 | 0.81 |
| $t_{50}$ | FN | 0.9 | 0.87 | 0.89 | 0.9 |
| | Meta | 0.9 | 0.88 | 0.89 | 0.9 |
| | TDGIA | 0.9 | 0.87 | 0.89 | 0.89 |
| | ours | 0.82 | 0.88 | 0.79 | 0.82 |

### B.4.4. RESULTS WITH LARGE-SCALE DATASET

To measure our approach on a larger dataset, we select the largest temporal graph dataset on the SNAP dataset collection(Leskovec & Sosič, 2016)—`Wiki-Talk-Temporal`(Paranjape et al., 2017)—for further analysis. This dataset represents Wikipedia users editing each other's Talk page. A directed edge $(u, v, t)$ means user $u$ edited $v$'s talk page at time $t$. **The graph has 1,140,149 nodes and 7,833,140 collected over 2320 days.**

The dataset has non-attributed edges, so we set them as all zero vectors. Note that we set the memory size to 64 instead of 172 to avoid the Out-Of-Memory issue. Due to the time limit, we train TGN and Roland for ten epochs instead of 20 in our prior experimental settings. The results are shown in Table 9. As we can observe, even for a very large graph with a 1% node budget, our attack shows a similar behavior as our prior results –Our attack is long-lasting and can affect more nodes' predictions in the future.

*Table 9.* Accumulated accuracy of edge prediction in the vanilla/attacked TGNNs over different timestamps on `Wiki-Talk-Temporal`.

| Dataset | | Wiki-Talk-Temporal | | | |
|---|---|---|---|---|---|
| Attack Budget | | 1% | | 5% | |
| Model | | TGN | ROLAND | TGN | ROLAND |
| **Vanilla** | | 0.97 | 0.98 | 0.97 | 0.98 |
| $t_0$ | FN | 0.89 | 0.90 | 0.83 | 0.88 |
| | ours | 0.94 | 0.91 | 0.86 | 0.88 |
| $t_{25}$ | FN | 0.98 | 0.97 | 0.97 | 0.96 |
| | ours | 0.92 | 0.90 | 0.82 | 0.89 |
| $t_{50}$ | FN | 0.97 | 0.98 | 0.97 | 0.97 |
| | ours | 0.91 | 0.91 | 0.84 | 0.86 |

## B.5. Extra results on attacks under defenses

We include the results of two attacks, i.e., FakeNode and MemFrezzing, under the two defenses, i.e., `adv_train` and `Lip_reg`, on two TGNN models, i.e., JODIE and Dyrep. The observations are similar to the prior analysis.

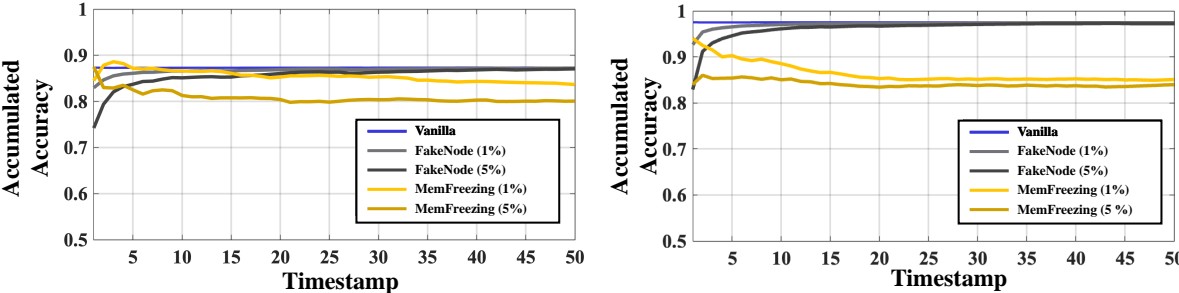

*Figure 11.* Accumulated accuracies of DyRep under `Adv_train`(left), and `Lip_reg`(right) with FakeNode and our attack on `WIKI`.

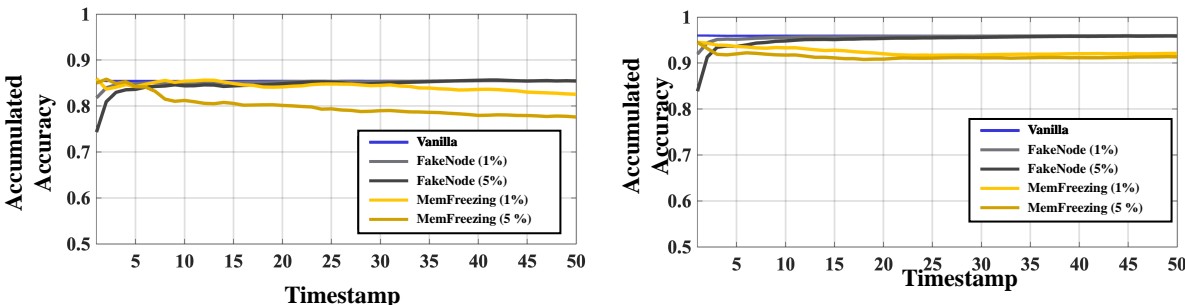

*Figure 12.* Accumulated accuracies of JODIE under `Adv_train`(left), and `Lip_reg`(right) with FakeNode and our attack on `WIKI`.

To give a more in-depth evaluation, we design a defending method by leveraging the data-filtering concept in GNN-Guard(Zhang & Zitnik, 2020) for the evasion attack. Specifically, following the insights that only the similar node may provide significant information for prediction, GNNGuard adopts a cosine-similarity-based approach to discount the messages passing between dissimilar nodes. So, we also use the cosine similarities to rank and filter the messages. Specifically, similar to the GNNGuard, we compute the similarities between two nodes. For each node, we normalize the similarities between it and its neighbors, then prune the lower 50% (same as GNNGuard). We show the experiment results in Table 10.

*Table 10.* Attack Performance under the **GNNGurad**.

| Attack Budget | | 1% | | | | 5% | | |
|---|---|---|---|---|---|---|---|---|
| **Dataset** | | **WIKI** | | **REDDIT** | | **WIKI** | | **REDDIT** |
| **Model** | | TGN | ROLAND | TGN | ROLAND | TGN | ROLAND | TGN | ROLAND |
| **Vanilla** | | 0.93 | 0.94 | 0.96 | 0.95 | 0.93 | 0.94 | 0.96 | 0.95 |
| **After defense Acc.** | | 0.92 | 0.91 | 0.94 | 0.90 | 0.92 | 0.91 | 0.94 | 0.90 |
| $t_0$ | FN | 0.87 | 0.81 | 0.9 | 0.84 | 0.82 | 0.86 | 0.82 | 0.81 |
| | ours | 0.87 | 0.88 | 0.91 | 0.88 | 0.85 | 0.83 | 0.8 | 0.82 |
| $t_{25}$ | FN | 0.9 | 0.91 | 0.93 | 0.9 | 0.91 | 0.91 | 0.92 | 0.9 |
| | ours | 0.84 | 0.87 | 0.82 | 0.81 | 0.79 | 0.81 | 0.81 | 0.8 |
| $t_{50}$ | FN | 0.92 | 0.91 | 0.94 | 0.9 | 0.92 | 0.91 | 0.93 | 0.9 |
| | ours | 0.83 | 0.85 | 0.81 | 0.81 | 0.76 | 0.82 | 0.8 | 0.83 |

## B.6. Extra results with gradually injected attacks

MemFreezing can be effective in both one-time and multiple-time attacks. We show the results of multiple-time attacks, in which attacks are injected right before $t_0, t_5, t_{10}, t_{15}$ with 1% attack budget (i.e., 1% of all nodes) each time. The results are shown in Table 13.

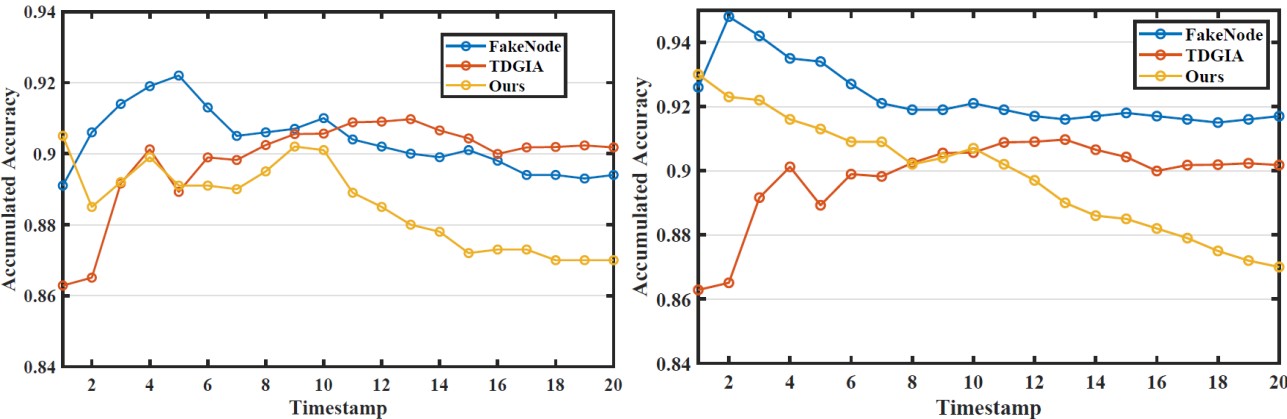

*Figure 13.* The accumulated accuracy under gradually injected attack to TGN on `WIKI`(left) and `REDDIT`(right). The attacks are injected right before $t_0, t_5, t_{10}, t_{15}$ with 1% attack budget (i.e., 1% of all nodes) each time.

As one can observe, with multiple attack times, MemFreezing effectively decreases accuracies, while the FakeNode and TDGIA attacks have shorter effective periods and fail to achieve similar accuracy drops. This is because the attack introduced by these baseline attacks will be weakened once there are changes between the graph at the attack and the prediction timestamp, and even multiple-time attacks cannot ensure that the attacks are just injected right before each prediction; in contrast, the noises from our attack can last over graph changes and even be boosted by future attacks.

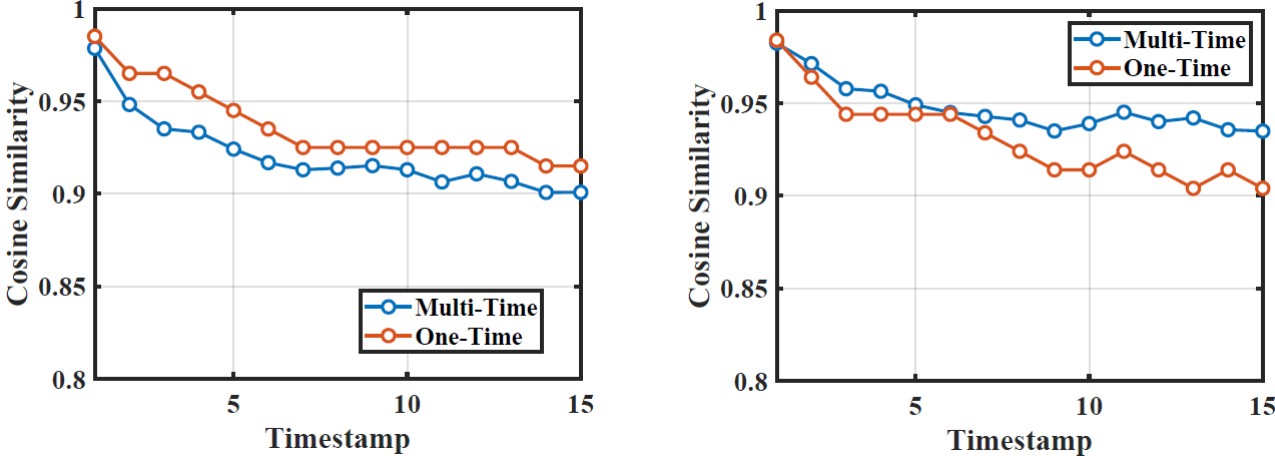

*Figure 14.* The similarities between victim nodes' initial noisy memories (at the time of the attack) and themselves'/their subsequent neighbors' memories in MemFreezing under one-time attack setup and multiple-times attack setup.

To evaluate the effectiveness of cross-freezing under multiple-time attack cases, we investigate the similarities between victim nodes' initial noisy memories (at the time of the attack) and themselves'/their subsequent neighbors' memories in MemFreezing under one-time attack setup and multiple-times attack setup (following the setup in Figure 7 in our paper). As shown in Figure 14, despite multiple times of injections, MemFreezing significantly raises the similarities between nodes' memories. The results demonstrate that the cross-freezing mechanism works effectively under multiple time attacks.

## B.7. Extra results on injecting attacks at different time stamps

To examine if MemFreezing can be effective despite the time of injection. We test its effectiveness under different injection timestamps instead of $t_0$, then evaluate its performance in the subsequent 50 timestamps. For instance, we may inject it at $t_{10}$ and then evaluate the accumulated accuracies in the original TGNN models and those under attack at $t_{50}$. The results of TGN on WIKI and REDDIT are shown in Figure 15. As one can observe, the attack effects remain similar despite its injecting time, demonstrating that MemFreezing can yield long-lasting and contagious attack at arbitrary attack time.

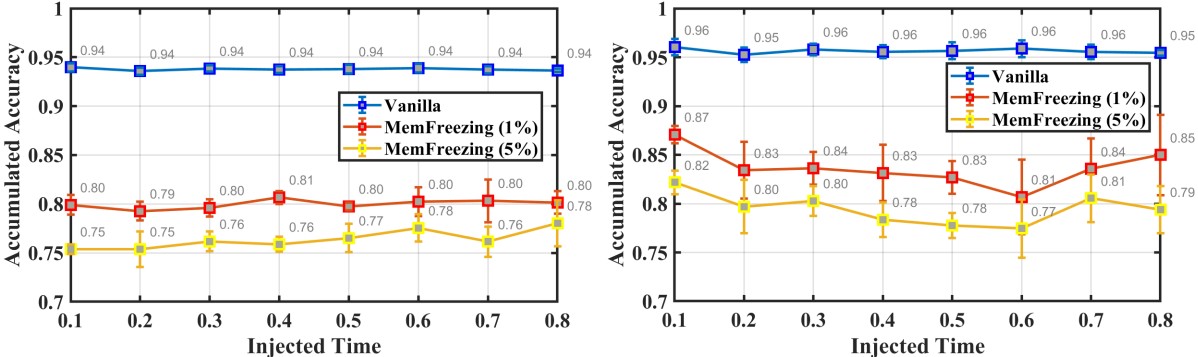

*Figure 15.* Accumulated accuracy under attack at various timestamps on TGN for WIKI (left) and REDDIT (right). Attacks are injected at 10%, 20%, 30%, 40%, 50%, 60%, 70%, and 80% of the total test set.

## B.8. Extra ablation study

We include the results for the ablation studies under the TGN model and REDDIT dataset in Figure 16. The results show a similar pattern as we observed in Section 5.

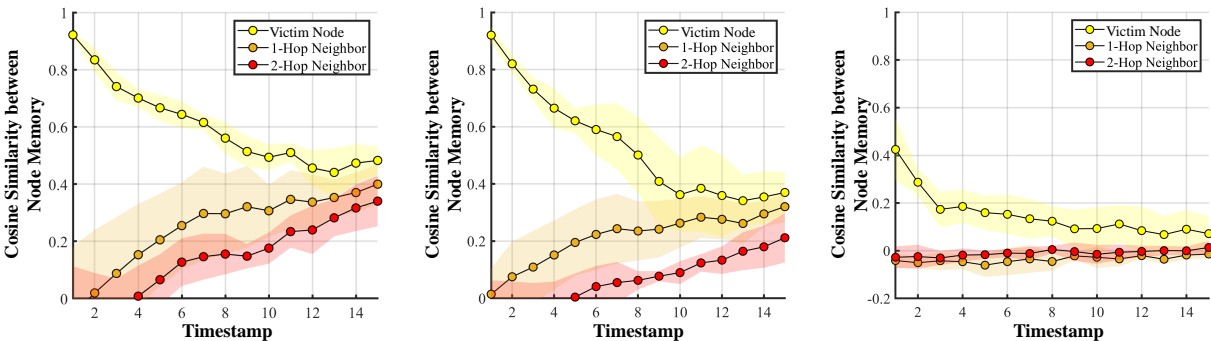

*Figure 16.* The similarities between victim nodes' initial noisy memories (at the time of the attack) and themselves'/their subsequent neighbors' memories in MemFreezing w/o (left) converge state, MemFreezing w/o freezing loss (middle), and regular nodes (right). All results are from the TGN model and REDDIT dataset.

## B.9. Analysis on freezing objective

To demonstrate the challenge of maximizing prediction losses, we add an extra term, $\mathcal{L}_u^{adv}$ to maximize the loss of predictions. Specifically, for each node $u$ we change Equation 7 in our paper as follows,

$$\mathcal{L}_u = \mathcal{L}_u^{freeze} + \mathcal{L}_u^{prop} - \gamma \cdot \mathcal{L}_u^{adv}$$

We use a coefficient $\gamma$ to control the ratio of adversarial losses. The adversarial loss $\mathcal{L}_u^{adv}$ is defined as follows,

$$\mathcal{L}_u^{adv} = \sum_i \ell(y_i, t_i) \mid i \in N(u)$$

In which $y_i$ presents the prediction result for the node $i$, $t_i$ is the ground truth of the prediction, and $\ell(y_i, t_i)$ indicates the binary-cross-entropy loss between them. Similar to baselines, for each node $u$, the objection function is to maximize the prediction loss of all its neighbors. We present the prediction accuracies under different $\gamma$ selections in Figure 17.

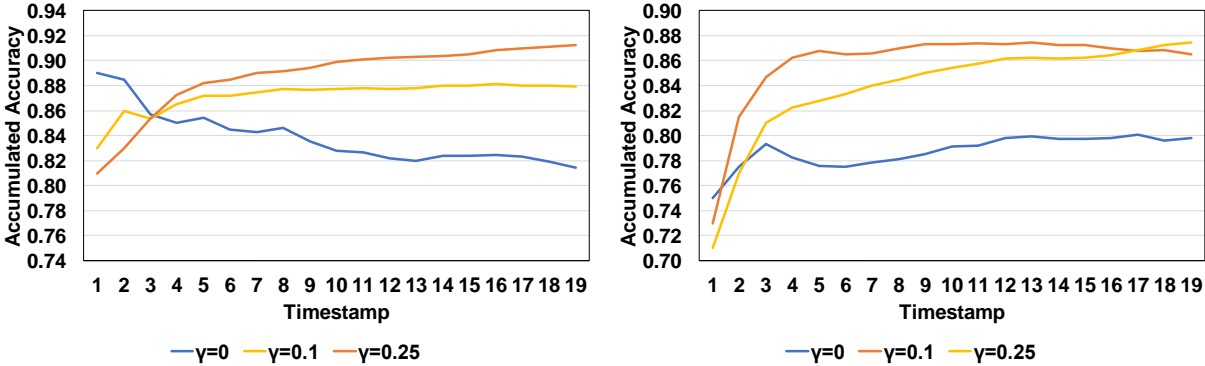

*Figure 17.* The accumulated accuracy with maximizing prediction losses under different $\gamma$ selections on TGN in WIKI(left) and REDDIT(right).

As shown in the figure, maximizing the adversarial losses can harm the predictions in the first batch (if and only if the predictions are made immediately after the attack). In the later batches, the effectiveness of the noise decreases drastically.

To further understand the reasons behind this, we investigate the similarities between victim nodes' initial noisy memories (at the time of the attack) and their memories in the future—termed as Persist Similarity—in Figure 18, the similarities between victim nodes' memories and their neighbors' memories—called Propagate Similarity in Figure 19.

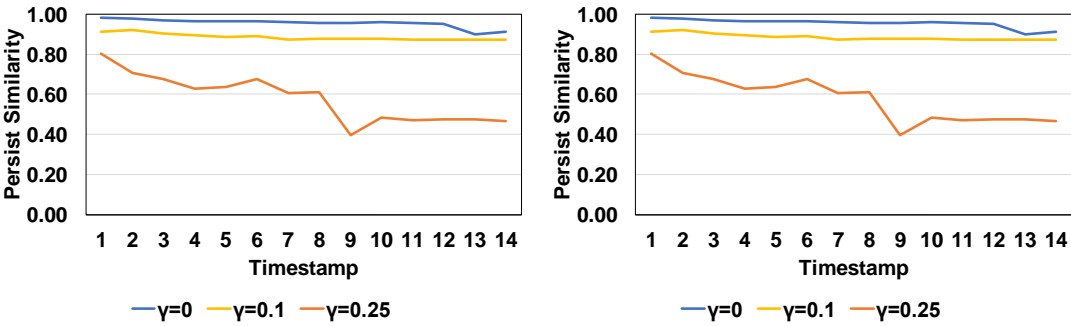

*Figure 18.* The similarities between victim nodes' initial noisy memories (at the time of the attack) and their memories in the future. The results are collected from TGN on WIKI(left) and REDDIT(right).

As one can observe, while introducing the adversarial losses, both persist and propagate similarities drop significantly, indicating that the nodes' memories cannot maintain the noisy states and may recover soon.

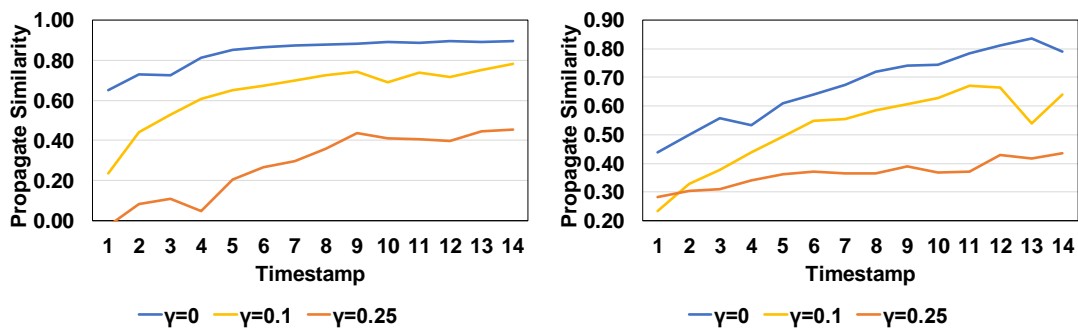

*Figure 19.* The similarities between victim nodes' memories and their future neighbors' memories. The results are collected from TGN on `WIKI`(left) and `REDDIT`(right).

### B.10. Stealthness analysis

As discussed in Appendix B.2, we select ranges of noisy messages between -1 and 1 since -1 and 1 are the theoretical minimum and maximum values of the clean messages. To further investigate if the MemFreezing attack introduces enough stealth fake events/nodes, we further investigate the range of message-wise means (i.e., means of all features over each message) and message-wise standard deviation (i.e., the standard deviation of all features over each message) for clean and noisy messages produced by different attacks in Table 11.

*Table 11.* Ranges of message-wise mean and standard deviation over all of the clean messages (Clean) and noisy messages produced by MemFreezing in `WIKI` and `REDDIT`.

|  | **WIKI** | | **REDDIT** | |
| --- | --- | --- | --- | --- |
|  | **Mean[min,max]** | **Std[min,max]** | **Mean[min,max]** | **Std[min,max]** |
| **Clean** | [-0.033, 0.106] | [0.206, 0.866] | [-0.093, 0.146] | [0.202, 0.789] |
| **MemFreezing** | [-0.014, 0.044] | [0.426, 0.570] | [-0.012, 0.038] | [0.580, 0.695] |
| **FakeNode** | [0.003 , 0.008 ] | [0.628 , 0.702] | [-0.030, 0.018] | [0.525, 0.686] |

The range of mean and std of our noisy messages are included within the range of those in the clean message and are similar to the baseline attack, demonstrating that their distributions or magnitudes are similar to the other features in the graph. Moreover, MemFreezing can effectively penetrate the defenses of GNNGuard, which uses similarity to filter susceptible messages in which the nodes/events with apparently different information (i.e., having low similarities compared to other nodes/events), as shown in Appendix B.5. In summary, the results indicate that MemFreezing can freeze node memories in TGNN without introducing significant different nodes/events that can be detected by existing GNN adversarial defenses.

### B.11. Extra sensitivity study

We include more results for different target node sampling strategies and attack budgets in Figure 20. The results show a similar pattern as we observed in Section 5.

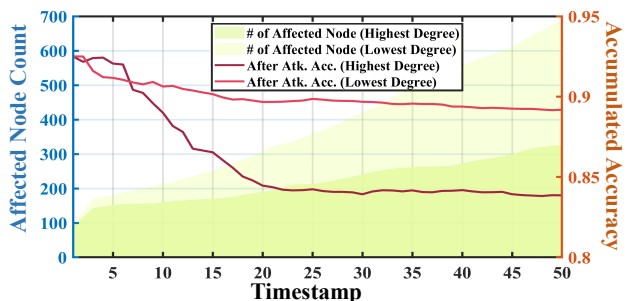 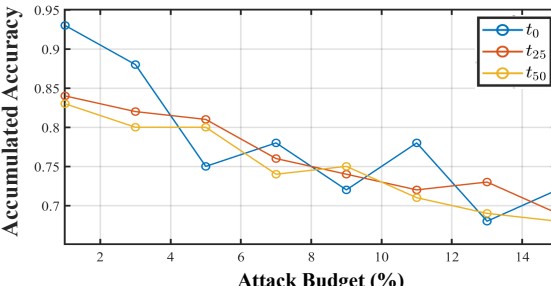

*Figure 20.* (left) Comparison between two strategies for selecting the injected node: lowest degree and highest degree nodes. Count of affected nodes and overall accuracy over time. (RIGHT) The accumulated accuracy at $t_0$, $t_{25}$, and $t_{50}$ under different attack budgets (% of total nodes). All results above are from TGN and REDDIT

## B.12. Accumulated Accuracies Over Time on Diverse Models

We report the accumulated accuracies over time collected from TGN, JODIE, and Dyrep on the `WIKI` and `REDDIT` datasets. The results include model accuracies under the vanilla (i.e., un-attacked), baseline (i.e., FakeNode), and our (i.e., MemFrezzing) attacks in edge prediction tasks.

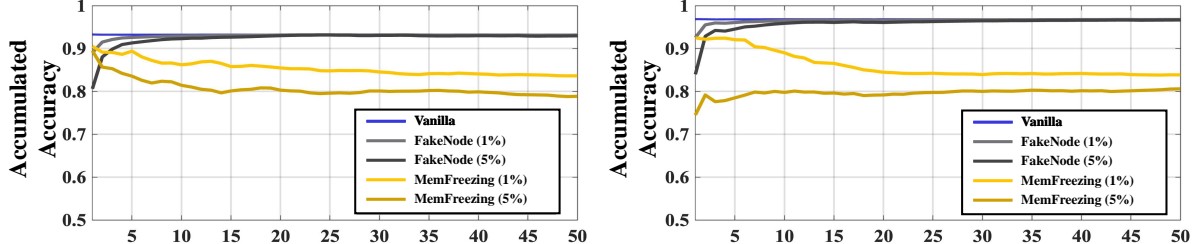

*Figure 21.* Accumulated accuracies of TGN under different attacks in link prediction tasks over time in `WIKI` (left) and `REDDIT` (right) datasets.

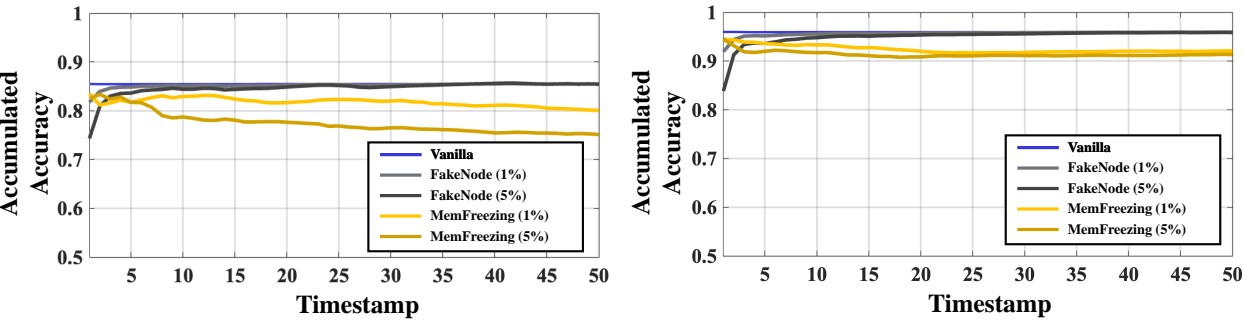

*Figure 22.* Accumulated accuracies of JODIE under different attacks in link predictions over time with `WIKI` (left) and `REDDIT` (right) datasets.

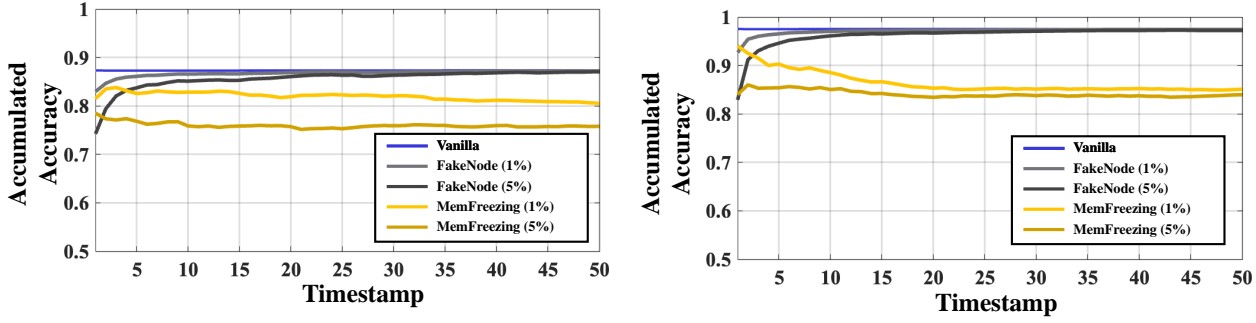

*Figure 23.* Accumulated accuracies of Dyrep under different attacks in link predictions over time with `WIKI` (left) and `REDDIT` (right) datasets.

## B.13. Affected Nodes

We report the number and accumulated accuracies over time of affected nodes over time in JODIE and Dyrep on the `WIKI` and `REDDIT` datasets. The results include model accuracies under our (i.e., MemFrezzing) attack in edge prediction tasks.

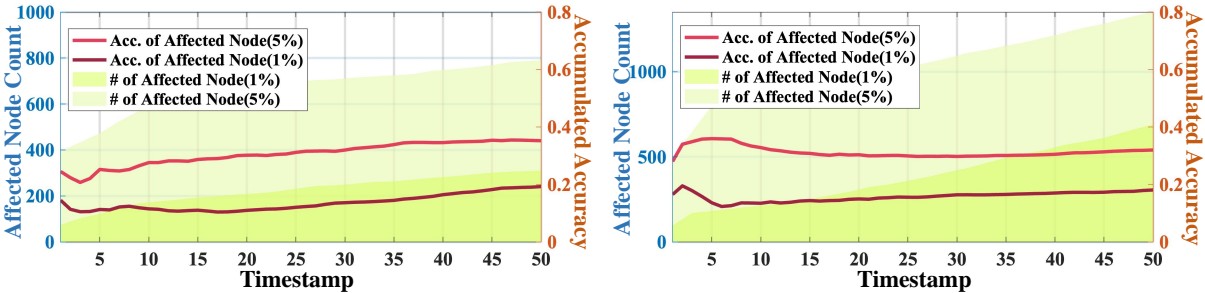

*Figure 24.* Count of affected nodes (presented as the colored areas) and their accumulated accuracies (presented as lines) in `WIKI` (left) and `REDDIT` (right) over time. The data are collected in TGN.

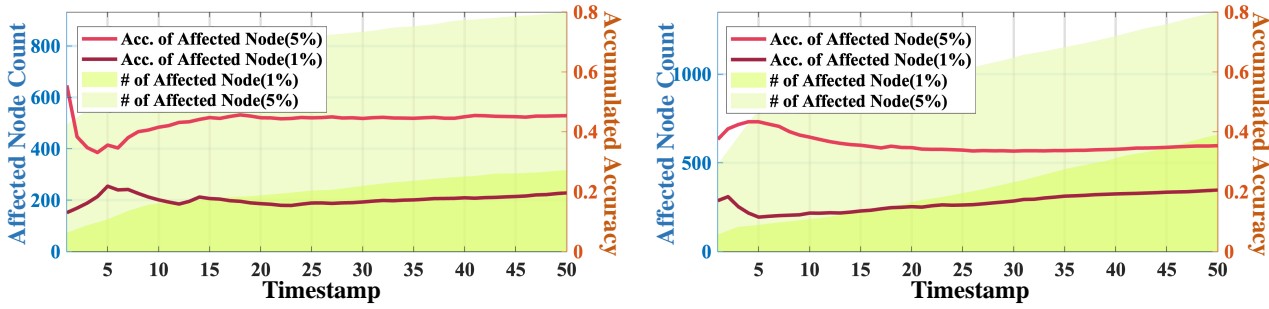

*Figure 25.* Count of affected nodes (presented as the colored areas) and their accumulated accuracies (presented as lines) in `WIKI` (left) and `REDDIT` (right) over time. The data are collected in JODIE.

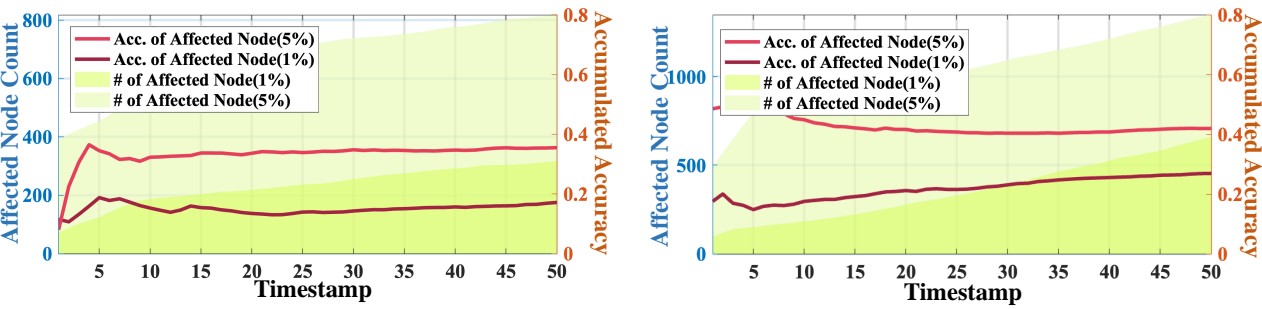

*Figure 26.* Count of affected nodes (presented as the colored areas) and their accumulated accuracies (presented as lines) in `WIKI` (left) and `REDDIT` (right) over time. The data are collected in Dyrep.

## B.14. Noise Propagating

We report the cosine similarities between the initial victim node and its neighbors over time in JODIE and Dyrep on the `WIKI` and `REDDIT` datasets. The results include similarities under our (i.e., MemFrezzing) attack in edge prediction tasks.

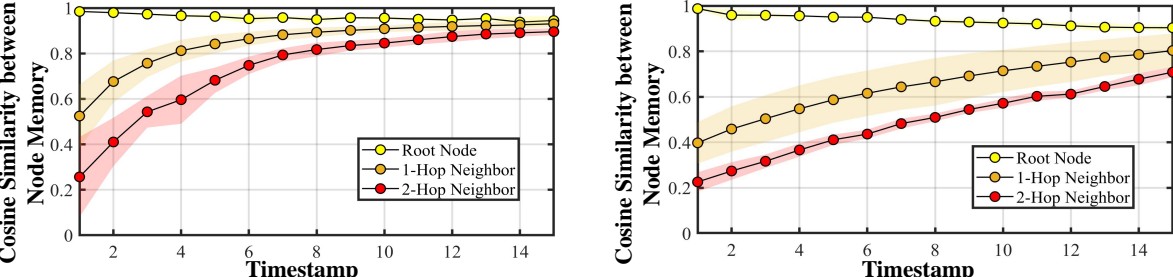

*Figure 27.* The similarities between victim nodes' initial noisy memories (at the time of the attack) and themselves'/their subsequent neighbors' memories in `WIKI` (left) and `REDDIT` (right) over time. The data are collected in TGN.

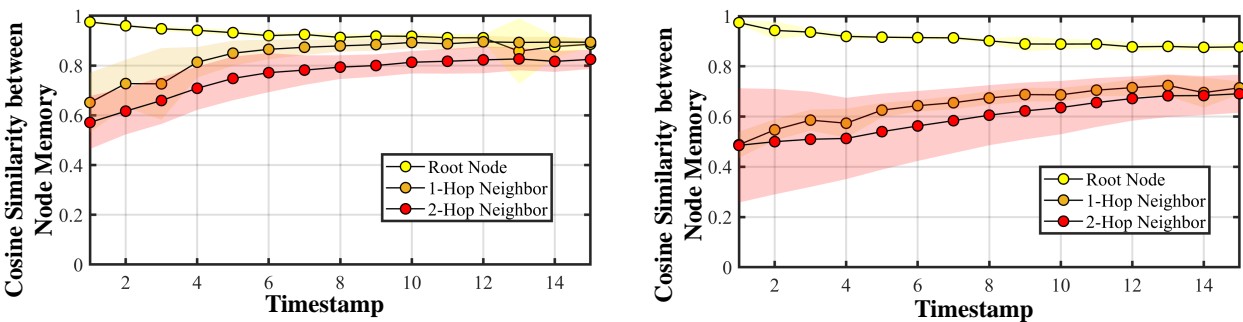

*Figure 28.* The cosine similarities between victim nodes' initial memory (at the time of the attack) and themselves/their subsequent neighbors' memories in `WIKI` (left) and `REDDIT` (right) over time. The data are collected in JODIE.

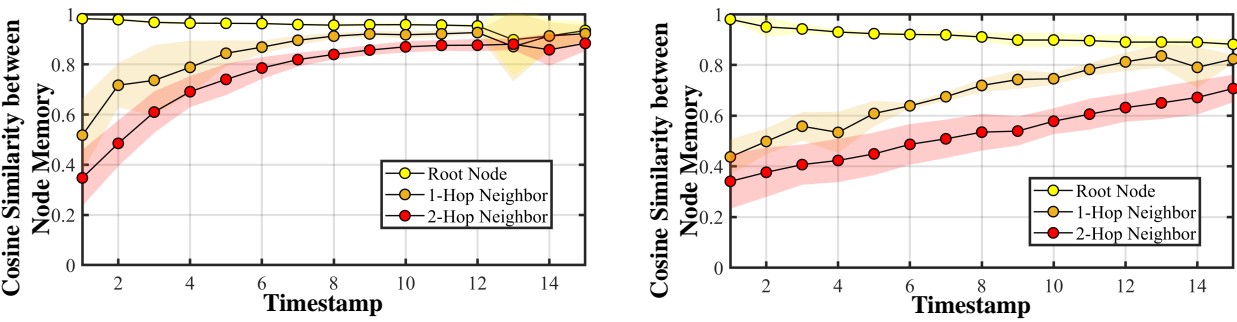

*Figure 29.* The cosine similarities between victim nodes' initial memory (at the time of the attack) and themselves/their subsequent neighbors' memories in `WIKI` (left) and `REDDIT` (right) over time. The data are collected in Dyrep.

## B.15. Evaluation on Attacks in Black-Box Setting

In the black-box setting, attackers do not have access to the target model's parameters. To this end, we evaluate black-box adversarial attacks using two commonly used setups: (i) surrogate models and (ii) zero-shot attacks.

### B.15.1. SURROGATE MODES

A common approach in the adversarial attack domain is to train a surrogate model locally and use it to generate adversarial examples that can transfer to the target model. For temporal graph datasets, many popular benchmarks such as Wiki and Reddit are open-source, providing well-labeled, diverse data annotated with detailed timestamp information. Hence, We first evaluate Memfreezing under the following setup: (1) We first train surrogate models on randomly sampled subsets (60% or 80%) of the original training data. (2) Then, we use these models to generate adversarial examples. (3) Lastly, we inject the generated adversarial examples into the dynamic graphs and evaluate the prediction accuracies in the target model trained on the complete (100%) dataset.

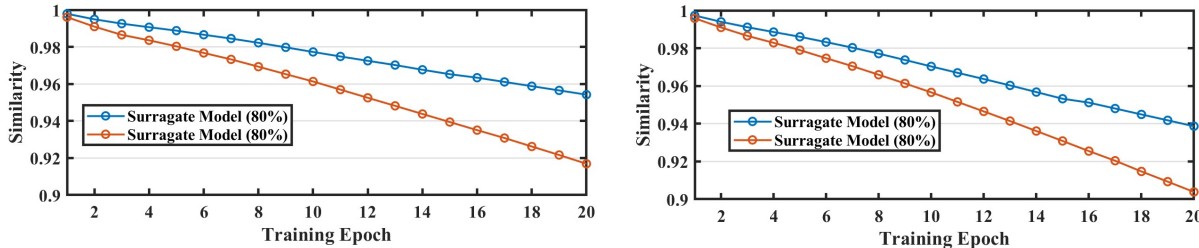

*Figure 30.* Cosine similarity of model parameters between surrogate model (80% and 60% training dataset) vs. target model. (left) are evaluated from TGN model using `WIKI` dataset; (right) are evaluated from TGN model using `Reddit` dataset.

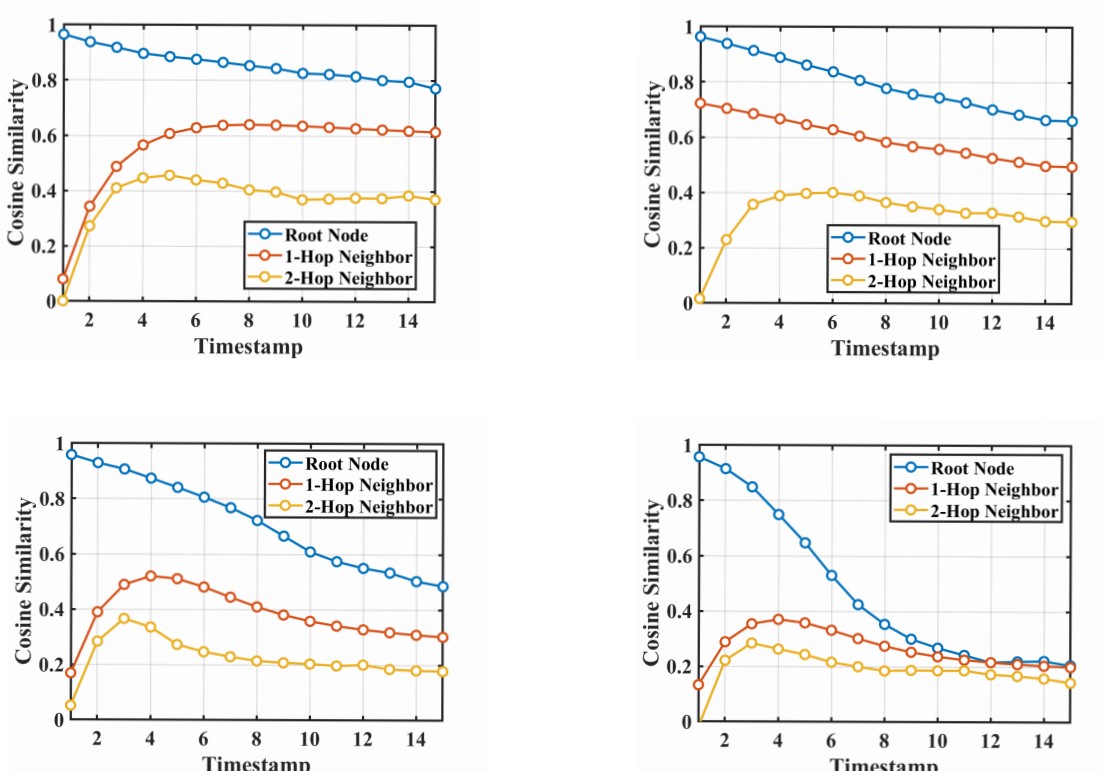

*Figure 31.* The similarities between victim nodes' initial noisy memories (at the time of the attack) and themselves'/their subsequent neighbors' memories. **The noisy patterns are computed from the surrogate model**. (Top-left) the surrogate model is trained with 80% dataset in `WIKI`; (Top-right) the surrogate model is trained with 80% dataset in `REDDIT`; (Bottom-left) the surrogate model is trained with 60% dataset in `WIKI`; (Bottom-right) the surrogate model is trained with 60% dataset in `REDDIT`.

We first investigate if surrogate models, which are trained with partial training data, can effectively represent the target models. In Figure 30, we analyze the similarity of parameters between the surrogate and target models. The analysis includes weights and biases from both the GRU units and the attention-based classifier. We track this similarity across training epochs 1 through 20. The results show a gradual decrease in similarity as training progresses. For the surrogate model trained on 80% of the dataset, the similarity decay is minimal, maintaining values between 0.94 and 0.96 at epoch 20. The surrogate model trained on 60% of the dataset shows a more pronounced decay, with similarity values ranging from 0.90 to 0.92 at epoch 20.

Next, we evaluate how effectively adversarial examples generated from surrogate models persist in target models. We first apply the MemFreezing algorithm to the surrogate model to generate noisy patterns, which we then transfer to the target model. We assess the attack's persistence by measuring the cosine similarity between the victim nodes' initial noisy memories (at the time of attacks) and both their own and their neighbors' memories over time. As shown in Figure 31, although the freezing capability is somewhat weakened due to the surrogate model's incomplete training data, our attack remains effective: the noisy patterns maintain a similarity above 0.8 even after 15 updates, and one-hop neighbors show significant influence with similarity values exceeding 0.6 relative to the noisy pattern.

Lastly, in Table 12, we compare MemFreezing against baseline attacks under the above-mentioned black-box setting by measuring their impact on overall model accuracy. While all attacks show reduced effectiveness in the black-box setting, MemFreezing maintains its superiority, achieving the largest accuracy drop compared to baseline attacks.

*Table 12.* Accumulated accuracy of edge prediction in the vanilla/attacked TGNNs over different timestamps on `REDDIT` and `WIKI`; The attack budget is 5% for all attacks; lower matrices indicate more effective attacks.

| Surrogate Model | Dataset | | WIKI | | | | REDDIT | | | |
|---|---|---|---|---|---|---|---|---|---|---|
| | Model | TGN | JODIE | Dyrep | ROLAND | TGN | JODIE | Dyrep | ROLAND |
| | **Vanilla** | 0.93 | 0.87 | 0.86 | 0.94 | 0.97 | 0.98 | 0.96 | 0.95 |
| 80% | $t_0$ | FN | 0.87 | 0.85 | 0.82 | 0.86 | 0.93 | 0.94 | 0.92 | 0.93 |
| | | Meta-h | 0.90 | **0.83** | **0.81** | **0.85** | 0.93 | 0.95 | 0.90 | 0.92 |
| | | TDGIA | **0.87** | 0.84 | 0.82 | 0.85 | **0.91** | **0.88** | **0.86** | **0.88** |
| | | ours | 0.90 | 0.85 | 0.86 | 0.88 | 0.92 | 0.91 | 0.95 | 0.91 |
| | $t_{25}$ | FN | 0.93 | 0.87 | 0.85 | 0.94 | 0.97 | 0.97 | 0.96 | 0.95 |
| | | Meta-h | 0.93 | 0.87 | 0.84 | 0.93 | 0.96 | 0.98 | 0.94 | 0.96 |
| | | TDGIA | 0.93 | 0.87 | 0.85 | 0.94 | 0.97 | 0.98 | 0.96 | 0.95 |
| | | ours | **0.83** | **0.83** | **0.83** | **0.84** | **0.89** | **0.90** | **0.92** | **0.88** |
| | $t_{50}$ | FN | 0.94 | 0.87 | 0.86 | 0.94 | 0.97 | 0.97 | 0.96 | 0.95 |
| | | Meta-h | 0.93 | 0.87 | 0.85 | 0.93 | 0.97 | 0.98 | 0.94 | 0.95 |
| | | TDGIA | 0.93 | 0.87 | 0.85 | 0.93 | 0.96 | 0.97 | 0.95 | 0.92 |
| | | ours | **0.84** | **0.85** | **0.82** | **0.85** | **0.90** | **0.91** | **0.93** | **0.90** |
| 60% | $t_0$ | FN | 0.91 | 0.86 | 0.86 | 0.92 | 0.96 | 0.96 | 0.93 | 0.94 |
| | | Meta-h | **0.90** | **0.83** | **0.81** | **0.85** | 0.93 | 0.95 | 0.90 | 0.92 |
| | | TDGIA | 0.90 | 0.84 | 0.82 | 0.89 | **0.92** | **0.90** | **0.90** | **0.92** |
| | | ours | 0.92 | 0.87 | 0.85 | 0.90 | 0.93 | 0.93 | 0.92 | 0.93 |
| | $t_{25}$ | FN | 0.93 | 0.87 | 0.85 | 0.94 | 0.97 | 0.97 | 0.96 | 0.95 |
| | | Meta-h | 0.93 | 0.87 | 0.84 | 0.93 | 0.96 | 0.98 | 0.94 | 0.96 |
| | | TDGIA | 0.93 | 0.87 | 0.84 | 0.94 | 0.97 | 0.98 | 0.96 | 0.95 |
| | | ours | **0.87** | **0.84** | **0.84** | **0.87** | **0.90** | **0.91** | **0.93** | **0.91** |
| | $t_{50}$ | FN | 0.94 | 0.87 | 0.86 | 0.94 | 0.97 | 0.97 | 0.96 | 0.95 |
| | | Meta-h | 0.93 | 0.87 | 0.85 | 0.93 | 0.97 | 0.98 | 0.94 | 0.95 |
| | | TDGIA | 0.93 | 0.87 | 0.86 | 0.93 | 0.96 | 0.97 | 0.95 | 0.95 |
| | | ours | **0.87** | **0.86** | **0.85** | **0.90** | **0.92** | **0.92** | **0.94** | **0.90** |

### B.15.2. ZERO-SHOT ATTACKS

It is also possible that, in real-world cases, the attackers have no idea about the model architecture or training datasets. To this end, we evaluate Memfreezing under the zero-shot transfer attack setup following the ensemble-based approach first

proposed in (Liu et al., 2016). Specifically, on the Wikipedia dataset, we generate the fake message by jointly optimizing the adversarial message for three models and then evaluating the effectiveness of this unified adversarial message on diverse models. As shown in 32, although it performs worse than the white-box attack, Memfreezing can still effectively perturb model predictions. We also acknowledge that memfreezing is less harmful than cases with more accurate model information.

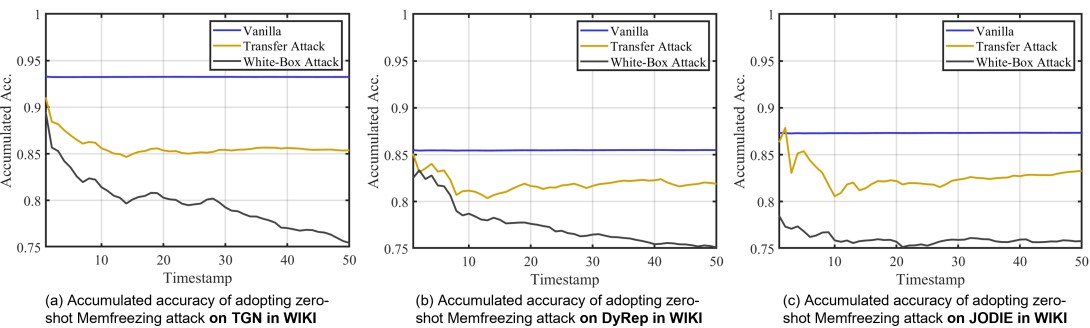

(a) Accumulated accuracy of adopting zero-shot Memfreezing attack **on TGN in WIKI**

(b) Accumulated accuracy of adopting zero-shot Memfreezing attack **on DyRep in WIKI**

(c) Accumulated accuracy of adopting zero-shot Memfreezing attack **on JODIE in WIKI**

*Figure 32.* The comparisons between performances of Memfreezing under zero-shot attack setup and white-box counterparts.

## B.16. Analysis on Future Simulation

To further understand if using nodes' current neighbor can be effective in extremely irregular and random graphs, we conduct the following experiments.

First, we further analyze the similarity among nodes' neighbors in diverse datasets using the same setup (e.g., model) as Figure 3(d). As shown in Figure 33, generally, nodes tend to have similar neighbors across diverse datasets. Hence, using current neighbors reasonably approximates future graph changes in practice.

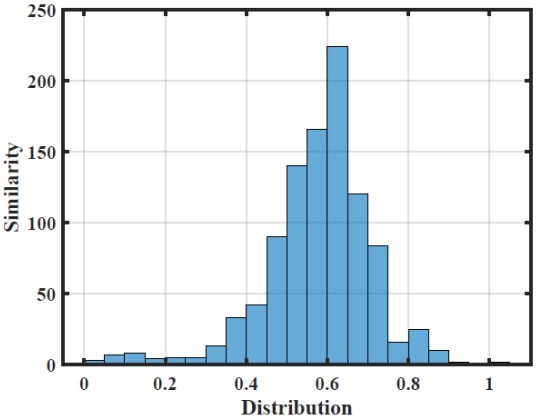 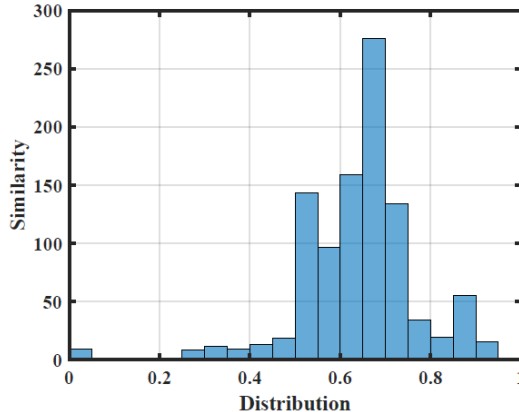

*Figure 33.* The distribution of cosine similarities among the ideal frozen states in different nodes in `REDDIT` and `REDDIT-BODY` datasets.

To investigate if our future neighbor simulation scheme is sufficient to freeze neighbors under irregular or highly random dynamic graphs, we simulate an irregular and random graph on top of the Wikipedia dataset. Specifically, we have victim nodes in the graph connected to nodes with random memories in the future timestamps. We also explored an alternative scheme to investigate whether the heuristic could be further enhanced. Specifically, in this alternative, we simulate nodes' future neighbors using nodes with random memories.

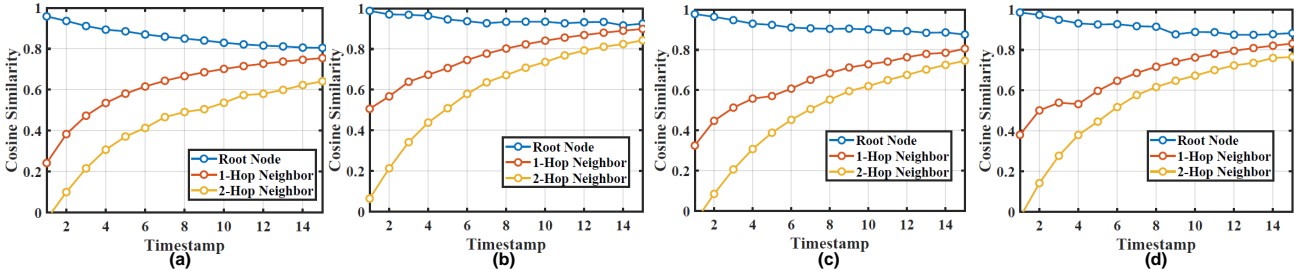

*Figure 34.* The similarities between victim nodes' initial noisy memories (at the time of the attack) and themselves'/their subsequent neighbors' memories in `WIKI` dataset and its randomized version under vanilla cases including (a) Using current neighbor for simulation under a noisy future, (b) Using current neighbor for simulation under a normal future, (c) Using random memory neighbor for simulation under a noisy future, (d) Using random memory neighbor for simulation under a normal future.

As shown in Figure 34(a), although resulting in lower similarities, MemFreezing effectively freezes these random neighbors (as shown in (a)). This demonstrates that our future simulation schemes are effective under even (i.e., Current Simulation) in irregular setups. The reason behind this is that, in addition to using current neighbors, we also simulate "new future neighbors" with all-zero memories, which further enhance the noise's capability to freeze unseen nodes.

Although the alternative scheme (i.e., Random Simulation) performs better under random neighbor cases (i.e., Noise Future), as shown in Figure 34(c); it shows worse performances in the real cases (i.e., Normal Future), as shown in Figure 34(d) compared to as shown in Figure 34(b). These findings collectively suggest that using current neighbors as surrogates is both practical and effective, even in challenging dynamic graph scenarios.

### B.17. Effective in LSTM-based TGNNs

While existing TGNN uses RNN and GRU for node memory updating (Rossi et al., 2020; Trivedi et al., 2019; Kumar et al., 2019; You et al., 2022), it is valuable to understand how nodes' memory is frozen under a memory updater with different RNN-variant.

To evaluate the effectiveness of MemFreezing when using LSTM as the memory updater, we replaced the GRU and RNN components in TGN (Rossi et al., 2020) with LSTM. We then assessed the performance of MemFreezing and baseline attacks under this new configuration. It is worth mentioning that since LSTM has two memories (i.e., long and short terms), they are different from GRU and RNN used in existing TGNNs. To adapt these two memories into one node memory under existing TGNN frameworks, we concatenate the two memories of a node together as its memory and freeze them altogether.

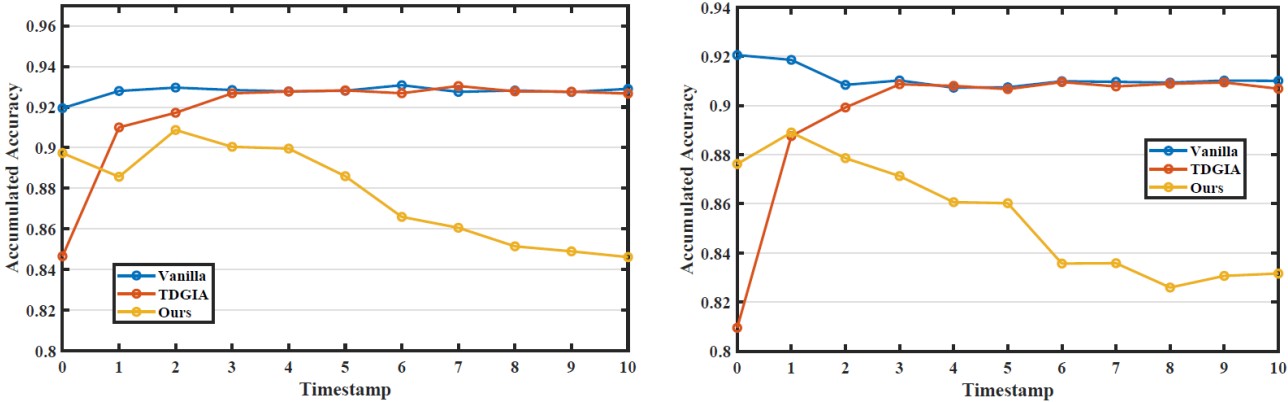

*Figure 35.* The accumulated accuracy of LSTM-based TGN under no-attack, TDGIA, and MemFreezing on `WIKI` (left) and `REDDIT` (right) datasets.

We first investigate the resulting accumulated accuracies in TGN. As shown in Figure 35, the LSTM-based TGN shows better robustness against MemFreezing. However, MemFreezing still effectively compromises predictions of LSTM-based TGN, leading to an average of 8% accuracy drops at $t_{100}$. In contrast, the baseline (i.e., TDGIA) still fails to disturb the predictions under limited-knowledge setups.

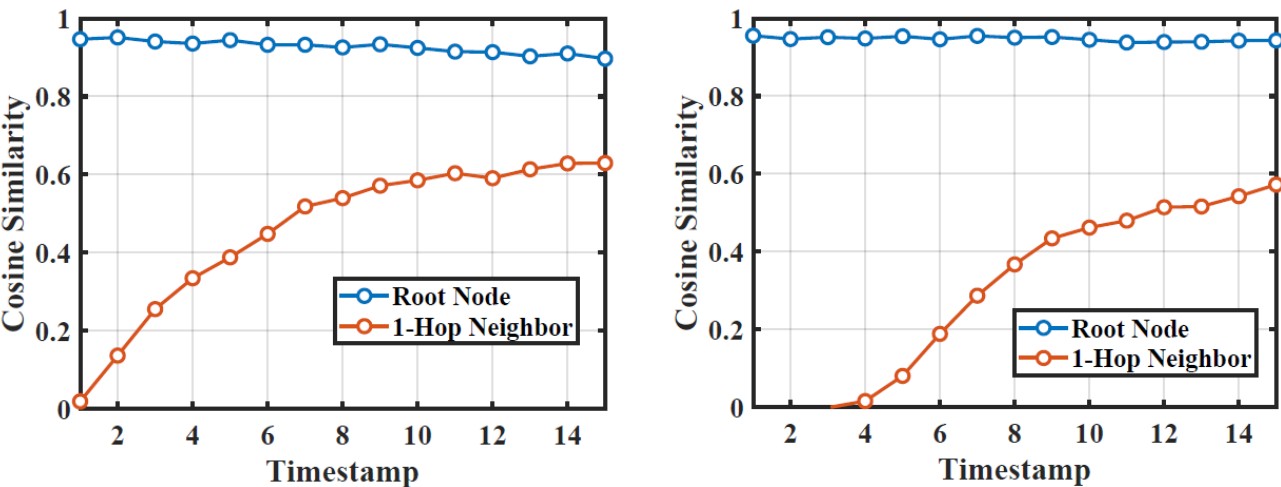

*Figure 36.* The similarities between victim nodes' initial noisy memories (at the time of the attack) and themselves'/their subsequent neighbors' memories in LSTM-based TGN on the `WIKI` dataset.

The LSTM-based TGN makes it more challenging since the attack has to freeze both long-term and short-term memories.

To understand the phenomenon, we further investigate the similarities between the victim nodes' initial memory and its subsequent and 1-hop neighbors' memories. As shown in Figure 36., the similarities between the victim nodes and their 1-hop neighbors are as low as around 0.6, which is not as high as the cases with GRU/RNNs (e.g., over 0.8).

# C. Discussion And Future Work

## C.1. Limits under different models and graphs.

While the experiment results in Appendix B.4 and Appendix B.11 demonstrated that MemFrezzing can be well-generalized on various inputs, several limitations can be observed according to the performance variance between different models. While our approach can effectively mislead TGN, ROLAND, and DyRep, its effectiveness is less significant on JODIE, which uses differences between a node's current and its last update time to decay the memory. From these observations, we deduce that our attack may encounter limitations in two specific scenarios:

- *Limited Influence of Node Memory on Predictions:* Our attack's effectiveness may be mitigated in situations where the node memory has a relatively minor role in influencing the model's predictions.

- *Usage of Additional Information in TGNN Models*: The effectiveness may also be constrained when the targeted TGNN model incorporates additional information beyond the node memory for its predictive processes.

While our attack strategy outperforms the baselines, these insights highlight potential limitations under certain model-specific conditions.

Nevertheless, Detecting a MemFreezing attack by observing node memory is challenging because nodes can naturally exhibit stable updates. For example, using TGN on the Wikipedia dataset, over 70% of node updates show high similarity (although they may not be consistently stable), which can also occur in real-world cases (e.g., an Amazon user with consistent shopping preferences). Thus, it is hard to differentiate an attacked node from naturally stable ones.

For instance, one could devise a "memory reset" module that resets the victim nodes' memories. To explore the potential of randomly and periodically resetting node memories, we conduct experiments in which we (a) randomly reset node memories upon each update or (b) reset memories after each 25 timestamps. As the results shown in Figure 37, doing so may jeopardize the models' clean accuracy with limited effectiveness in defending against Memfreezing.

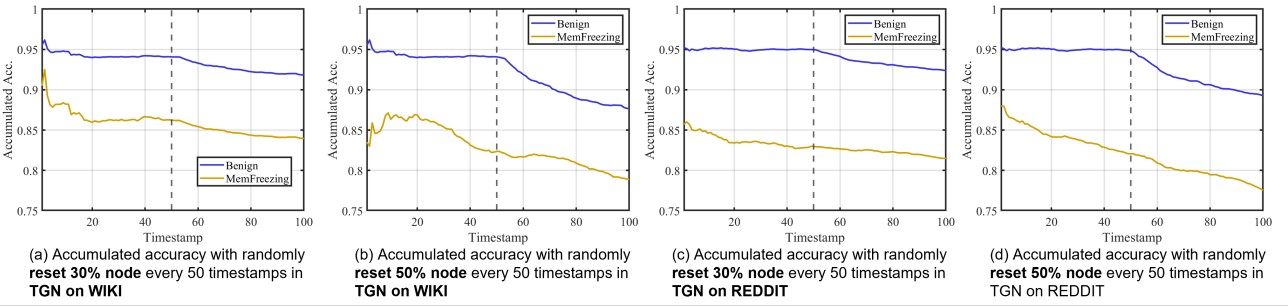

*Figure 37.* The performances of Memfreezing upon randomly resetting node memories.

## C.2. Potential Defenses.

While we demonstrate that many existing defense schemes, such as adversarial training or regularization, are less effective on our attacks, we expect a potential attack-oriented defense scheme for our attack using memory filtering. Specifically, a potential defensive approach for our attack is to pay less attention to the nodes' memory and rely more on their current input adaptively.

This scheme stems from the observation that our attacks are less effective on JODIE in node classification tasks. One key difference in JODIE is that it decays the node memory based on the time differences between the prediction time and the node's last update time. This mechanism introduces more hints (i.e., time differences) in addition to the memory itself, which cannot be effectively distorted by the attacks and yields some crucial information. For example, a Wikipedia user is less likely to be banned if he/she makes a new post after being inactive for a long while.

Therefore, using this non-memory information or current information that does not interact with node memory could

effectively hinder adversarial noises. To this end, an intelligent defense mechanism can judiciously filter out the memory and adaptively focus more on non-memory information if the memory is suspicious or potentially noisy.

## D. Complexity and Overheads of The Memfreezing Attack

We further approximate the time complexity of the MemFreezing, as it is crucial to understand its practicality. The time complexity of MemFreezing is approximately $\mathcal{O}(V + VD)$, where $V$ is the number of victim nodes being attacked and $D$ is their average degree.

The computation of MemFreezing can be divided into three main parts:

1. **Finding the Stable State**: For each victim node in $V$, we iteratively update its state using its two support neighbors until reaching the ideal stable state. Assuming a constant number of iterations for convergence, this step incurs a time complexity of $\mathcal{O}(V)$.

2. **Solving the Target Memory Using SGD**: For each victim node, we optimize the target memory state using stochastic gradient descent (SGD), considering (a) The node itself, (b) Its two support neighbors and (c)Its augmented neighbors. The total set has a size of at most $D + 20$ (current neighbors plus simulated neighbors), where $D$ approximates the number of the node's current neighbors. This optimization incurs a cost of $\mathcal{O}(D)$ per node, leading to a total time complexity of $\mathcal{O}(VD)$ across $V$ victim nodes with $D$ average degree.

3. **Introducing Fake Neighbors**: For each victim node, we compute and inject a fake neighbor to introduce noise. This step has a cost of $\mathcal{O}(1)$ per node, resulting in $\mathcal{O}(V)$ overall.

In summary, the overall time complexity of MemFreezing is dominated by the SGD optimization step for getting noisy memory, resulting in $\mathcal{O}(V + VD)$ time complexity. Under the worst cases, in which $D = V$ (e.g., fully connected graph), the complexity is $\mathcal{O}(V^2)$ We further show the comparisons of each attack's average latency per node in Table 13. The results show that the Memfreezing attack freezes nodes within seconds, indicating its potential in an online attack setup.

*Table 13.* Average latency per node of four attack methods on WIKI and REDDIT datasets.

| Dataset | Model | MemFreeze (s) | Meta-h ($\mu$s) | TDGIA (s) | FakeNode (s) |
|---------|-------|---------------|-----------------|-----------|--------------|
| WIKI | TGN | 0.48 | 546.88 | 1.91 | 0.87 |
| | Jodie | 0.49 | 957.51 | 1.63 | 0.81 |
| | Dyrep | 0.45 | 352.00 | 2.10 | 0.91 |
| | Roland | 0.53 | 816.78 | 3.28 | 1.27 |
| REDDIT | TGN | 0.66 | 262.63 | 1.46 | 0.81 |
| | Jodie | 0.46 | 158.91 | 1.49 | 0.76 |
| | Dyrep | 0.45 | 155.96 | 1.20 | 0.89 |
| | Roland | 0.53 | 392.90 | 2.14 | 1.47 |

