# OpenReview forum: "MemFreezing: A Novel Adversarial Attack on Temporal Graph Neural Networks under Limited Future Knowledge"
_ICML.cc/2025/Conference — ICML 2025 poster_

### Official Review · Reviewer_wVV6 · 2025-03-13

**Overall Recommendation:** 3

**Summary:**

The authors propose MemFreezing, an adversarial attack on temporal graph neural networks (TGNNs) that poisons TGNNs' recurrent neural network memory without knowledge about the future. For this, MemFreezing injects fake nodes into the graph that put their connected nodes into a "fronzen" state. Which means that their state becomes irresponsive and meaningless. The authors evaluate their attack on four TGNNs and compare to three baseline attacks.

**Claims And Evidence:**

The authors' claims are well supported by evidence.

**Essential References Not Discussed:**

The authors cover the related literature well.

What comes to my mind is the fact that the authors also choose a subset of nodes they aim to attack and adversarial training. Selecting the nodes that shall be attacked is a common problem in (static) graph robustness due to the simultaneous predictions on many nodes/edges. In other words, an attacker with a limited budget needs to decide on which nodes they are going to attack. While MemFreezing heuristically targets high-degree nodes, other works have, e.g., [1,2] discuss countermeasures/loss choices s.t. the attack optimization decides which nodes are to be attacked. Adversarial training and the important considerations were discussed previously for static graphs (e.g., [3,4]). Incorporating brief discussions could provide valuable pointers for future work.

[1] Ma et al. "Towards More Practical Adversarial Attacks on Graph Neural Networks" 2020

[2] Geisler et al. "Robustness of Graph Neural Networks at Scale" 2021

[3] Xu et al. "Topology Attack and Defense for Graph Neural Networks: An Optimization Perspective" 2019

[4] Gosh et al. "Adversarial Training for Graph Neural Networks" 2023

**Experimental Designs Or Analyses:**

The experimental design is sound. The experiments are comprehensive and also covers auxiliary aspects (defenses, stealthiness, …).

**Methods And Evaluation Criteria:**

Yes.

**Other Comments Or Suggestions:**

1. The last sentence before section 4.1 seems broken
1. $msg$ reads like $m \cdot s \cdot g$. I would recommend using "operatorname" or similar $\operatorname{msg}$. Same is true for UPDT, AGGR, GNN in Eq. 2-4
1. White space missing in Section B.3 after "GNNGuard"
1. Table 13 in Section B.6 should probably be Figure 13

**Other Strengths And Weaknesses:**

1. I am not particularly convinced that interesting all perturbations at a single timestamp is "realistic" nor desirable. I am much in favor of the setting where the attack is spread out over the duration of the test set. Hence, I would encourage the authors to place the results from section B.6 in the main body such that it is not overlooked.
1. It is not impossible that an adversary can have (limited) knowledge about the future. In some parts, it reads like this was not possible.
1. It might not be clear why vulnerabilities could be overlooked in idealized settings (e.g., as mentioned abstract). After all, the idealized setting is strictly more powerful. However, I agree that an attack in an idealized setting might not focus on vulnerabilities that are easy to utilize with limited knowledge. It seems that the attack is not generally revealing vulnerabilities in a more limited setting, it is rather utlizing one volnerabilitiy that exists in TGNN. Nevertheless, demonstrating that this limitation exists in TGNNs is a good contribution.
1. The authors tempered with the template to remove, e.g., "Anonymous Authors1" below the title and line numbers are missing.

**Questions For Authors:**

1. Why can adversaries reconstruct graphs of Meta or X reasonably well? Don't they have strict rate limits, effectively hindering crawling their data?
1. What perturbations are used for applying adversarial training? Is adversarial training applied to the training graph only?

**Relation To Broader Scientific Literature:**

The authors are the first to study adversarial attacks on TGNNs without knowledge about the future where they aim to put the memory in a frozen state. The authors reveal an important failure mode of TGNNs, and this failure mode is of interest to everyone who deploys TGNNs in practice.

**Theoretical Claims:**

Not applicable.

---

> ### Author Rebuttal · Authors · 2025-04-01
>
> **We sincerely thank the reviewer for the positive feedback and valuable comments. In response, we added more discussion on data crawling, adversarial training, and victim node selection, and we will also revise our paper accordingly.**
>
> ---
>
>  ## **Q1. Data Crawling**
> Thank you for the valuable question. While social media platforms like X or Meta indeed impose rate limits on their APIs, adversaries can bypass these restrictions using multiple accounts, rotating IP addresses (proxies), or publicly accessible endpoints such as profile pages. Furthermore, partial data from archives or aggregated third-party services can fill in missing pieces. These combined methods enable attackers (or researchers) to assemble sufficient public information on connections and interactions, yielding a reasonably accurate reconstruction of Meta or X’s dynamic graph, despite official rate limits.
>
> ---
>
>  ## **Q2. Adversarial training**
>
> We adopt a minimax scheme to conduct the adversarial training: the objective function can be formulated as:
>
> $$\min_\theta \max_\epsilon \mathcal{L} (f_\theta(s+\epsilon),y)$$
>
> We first perturb the memory of every training node by adding noise and use the noise to train the model. The training is conducted via a 10-step Projected Gradient Descent (PGD), and the noise is set to $\pm 0.2$.
>
>
> Although adversarial attacks and importance weighting have been extensively studied on static graphs [9, 10], these methods prove less effective on dynamic graphs for two main reasons:
>
> 1. **Dependence on a Complete Adjacency Matrix:**
>    Methods such as the min-max topology attack [9] and robust diffusion [10] rely on the full adjacency matrix \(A\) of the graph. However, the complete \(A\) is unknown in dynamic graphs because future edges and nodes are not seen at the training graph.
>
> 2. **Different Attack Objectives:**
>    Traditional adversarial training schemes target attacks that maximize classification or link prediction loss. In contrast, MemFreezing aims to freeze node memories into stable states, indirectly disrupting predictions. Since this mechanism does not directly alter the output loss, defenses designed for loss-based attacks are insufficient to counter MemFreezing.
>
>
> [9] Xu et al. "Topology Attack and Defense for Graph Neural Networks: An Optimization Perspective" 2019
>
> [10] Gosh et al. "Adversarial Training for Graph Neural Networks" 2023
>
> ---
>
>  ## **Q3. Attack node selection**
>
> We agree that selecting victim nodes under a limited attack budget is crucial and that more sophisticated methods beyond our high-degree heuristic may yield improvements. Our choice was motivated by the challenge of identifying optimal targets without precise future knowledge, as approaches in [12, 13] typically rely on stable graph structures or additional information about upcoming changes. In a dynamic setting, a node's importance (or receptive field) can shift drastically once it gains new neighbors, making future-oriented selection inherently difficult.
>
> Nonetheless, we acknowledge the potential for more specialized node selection methods, even under partial knowledge. To explore this, we integrated the importance-score-based node selection strategies from [11] into MemFreezing—applying their selection logic based solely on data available at the attack time—and tested both one-time and multiple-time attacks. As summarized in **Figure R6 (anonymous link:https://ibb.co/8LW5rtRc)**,  the scheme rates each node via an importance score, and the attack targets the node with the highest score. It achieves similar overall attack effectiveness compared to our simpler high-degree heuristic. This outcome suggests that, in highly dynamic scenarios, high-degree targeting remains a practical fallback, though we believe further research on adaptive selection in dynamic graphs is warranted.
>
> [11] Ma et al. "Towards More Practical Adversarial Attacks on Graph Neural Networks" 2020
>
> [12] Geisler et al. "Robustness of Graph Neural Networks at Scale" 2021

---

### Official Review · Reviewer_BqbP · 2025-03-13

**Overall Recommendation:** 3

**Summary:**

The paper studies adversarial attacks on temporal graph neural networks and proposes an effective approach to generate attacks that can persist over future timesteps. They consider an online adversarial attack setting and add fake nodes with carefully crafted memory representations at each timestep such that the victim nodes' memories reach an expected ``frozen'' state. Results on different temporal GNNs and tasks show the effectiveness of the MemFreezing attack as compared to baselines.

**Claims And Evidence:**

Yes

**Essential References Not Discussed:**

- Newer benchmark datasets should be considered for more comprehensive results [1].

- A discussion with other online adversarial attacks, as studied in the current work is also missing [2, Sharma et al., 2023].

- The paper studies node injection attacks in temporal graphs but does not discuss a similar setting studied in static graphs [3].

- The paper does not discuss a more recent dynamic graph attack that studies poisoning attacks but establishes various unnoticeability constraints.

[1] Huang, Shenyang, et al. "Temporal graph benchmark for machine learning on temporal graphs." Advances in Neural Information Processing Systems 36 (2023): 2056-2073.

[2] Mladenovic, Andjela, et al. "Online adversarial attacks." arXiv preprint arXiv:2103.02014 (2021).

[3] Chen, Yongqiang, et al. "Understanding and improving graph injection attack by promoting unnoticeability." arXiv preprint arXiv:2202.08057 (2022).

[4] Lee, Dongjin, Juho Lee, and Kijung Shin. "Spear and shield: adversarial attacks and defense methods for model-based link prediction on continuous-time dynamic graphs." Proceedings of the AAAI Conference on Artificial Intelligence. Vol. 38. No. 12. 2024.

**Experimental Designs Or Analyses:**

- The effectiveness of Memfreezing stagnates at a high perturbation rate with respect to accumulated accuracy while the premise of the work is to formulate attacks that can work regardless of future interactions.
- The proposed method does not work as well in the case of black-box setting, indicating that MSE between memories requires the models to be precise. Here, one should also study a simpler transferability setting where the attacks from one model are used for another.
- Running time analysis should also be conducted as this is supposed to happen in a practical online setting.
- It is not clear why the model omits Chen et al., 2021 and Sharma et al., 2023 for discrete graph models (ROLAND). In addition, they should also include temporal dynamics aware perturbation constraint for unnoticeability of these perturbations.

**Methods And Evaluation Criteria:**

- The paper claims more practicality for limited knowledge of the future graphs but gives full access to the model, which is also not well motivated.
- It is also not practical to inject edges with high-degree nodes as they can be easily detectable as malicious through simple anomaly detection methods. They should follow unnoticeability constraints like:
  - Lee, Dongjin, Juho Lee, and Kijung Shin. "Spear and shield: adversarial attacks and defense methods for model-based link prediction on continuous-time dynamic graphs." Proceedings of the AAAI Conference on Artificial Intelligence. Vol. 38. No. 12. 2024.
- The memory terminology is specific to certain kinds of models and it is not clear how it can be extended to other models that do not explicitly formulate past interactions of the nodes in memory representations such as:
  - Luo, Yuhong, and Pan Li. "Neighborhood-aware scalable temporal network representation learning." Learning on Graphs Conference. PMLR, 2022.
  - Besta, Maciej, et al. "Hot: Higher-order dynamic graph representation learning with efficient transformers." Learning on Graphs Conference. PMLR, 2024.
  - Wang, Yanbang, et al. "Inductive representation learning in temporal networks via causal anonymous walks." arXiv preprint arXiv:2101.05974 (2021).
- While the premise motivates that we do not know the timestep for the adversarial attack, the authors measure their task using a metric of accumulated accuracy which measures the accuracy at the current time step plus the previous timesteps accuracy. Thus, it basically measures the accuracy if the attack had happened at the current timestep. More detail and motivation about how the metric reflects the premise should be provided.

-

**Other Comments Or Suggestions:**

- The submission draft is not in the correct format.
- Impact statement is not included.

**Other Strengths And Weaknesses:**

- Motivating examples and insights are appreciated and supplement the writing well.
- A lot of main results are deferred to the Appendix which makes the it very hard to understand the key results.

**Questions For Authors:**

- How can the method be generalized to non-memory-based TGNNs?
- Why does the accumulated accuracy stagnate?
- How is it practical to form edges with high-degree nodes? Is it possible to make these attacks more unnoticeable given Spear and Shield and TDAP constraint?
- How do the perturbations transfer across victim models in order to test the attacks under the more practical black-box setting?

**Relation To Broader Scientific Literature:**

The paper studies the vulnerability of memory-based TGNNs in online evasion-based attacks.

**Theoretical Claims:**

N/A

---

> ### Author Rebuttal · Authors · 2025-04-01
>
> **We sincerely appreciate the valuable comments and insights from the reviewer. In response, we carefully respond to the reviewer’s questions and will also revise the paper accordingly.**
>
> ---
>
>
> ## **Q1. Generalizability to non-memory-based TGNNs**
>
> We acknowledge that MemFreezing primarily targets TGNNs that continuously track temporal information through evolving node features. Its memory-freezing objective is specifically designed to disrupt the dynamics inherent in such memory-based systems. For models that do not maintain temporal node features, the objective function needs to be modified. However, as recent studies [4] have shown, this TGNN family consistently achieves state-of-the-art performance on dynamic graph tasks, underscoring the practical relevance of our focus. We recognize the importance of extending our approach to non-memory-based architectures and plan to explore such adaptations in future work.
>
> ---
>
> ## **Q2. Discussion on accumulated accuracy**
>
> We use the accumulated accuracy to capture the impact of an attack over time—answering the question, “What is the accuracy of predictions up to a specific timestamp?” Specifically, accumulated accuracy is not equivalent to accuracy at a specific timestamp. Instead, it represents the accuracy up to a particular timestamp, and **the attack can happen anytime before/after the measurement**.
>
>  A **stagnant accumulated accuracy** indicates that new predictions continue to be misclassified, demonstrating that the adversarial effect persists over time. In contrast, if the model were recovering, the metric would gradually improve.
>
> ---
>
>  ## **Q3. Unnoticeability in targeting high-degree nodes**
>
> Thank you for the insightful question. As noted in [6] (Constraint C4), repeatedly injecting edges into a single high-degree node could trigger anomaly detection. However, MemFreezing introduces **only one fake edge per victim node**—even if that node is high-degree—thereby not violating C4 or exceeding typical per-node perturbation limits.
>
> Moreover, we explore if Memfreeze can integrate the TDAP constraints  [6]. For C1 and C4,  we follow our original setup to limit the number of changes (C1) and per node changes (C4). In addition, we enable C2 by restricting the number of changes per batch and enable C3 by only selecting victim nodes that have an event within the most recent five batches. The results in **Figure R4 (anonymous link: https://ibb.co/p6Mnw2hq)** indicate that Memfreezing can outperform the prior approach under such constraints, suggesting that it can be integrated with [6] to provide more unnoticeable attacks.
>
>
> [6] Lee, Dongjin, Juho Lee, and Kijung Shin. "Spear and shield: adversarial attacks and defense methods for model-based link prediction on continuous-time dynamic graphs." Proceedings of the AAAI Conference on Artificial Intelligence. Vol. 38. No. 12. 2024.
>
> ---
>
>  ## **Q4. Evaluation under transfer attacks**
>
> We agree that further exploring more black-box constraints is valuable. Please refer to our results in **Q2 to reviewer EZNr**.
>
> ---
>
>  ## **Q5. Runtime**
>
> We evaluate the average latency of Memfreezing and other attacks. The results in **Figure R5 (anonymous link:https://ibb.co/JwmqR2LD)** show each attack's average latency per node. The results show that the Memfreezing attack nodes within seconds, indicating its potential in an online attack setup.
>
> ---
>
>  ## **Q6. More related works**
>
> Thank you for suggesting additional literature. We note that the online adversarial attack in [7] supports our threat model by demonstrating attacks on streaming data rather than on fully known inputs. However, [7] differs from our work: even though it faces streaming data, predictions are made at the attack time with complete input information (e.g., an image), whereas in our setup, the input for prediction is unknown at the attack time, making the attack more challenging.
>
> Similarly, while the Homophily Unnoticeability approach in [8] effectively improves attack stealth, it relies on having the full adjacency matrix—a resource that is unavailable in our scenario. We analyze our attack's stealthiness in Appendix B10 and will expand our discussion on unnoticeability constraints from [8] and [6] in the revised paper.
>
> Lastly, we did not evaluate the methods in [Chen et al., 2021 and Sharma et al., 2023] because they assume a divergent threat model where attackers have full knowledge of the dynamic graph, allowing them to optimally select when and where to inject perturbations over the entire graph evolution. In our limited-knowledge scenario, determining the optimal allocation of the attack budget across timestamps is significantly more challenging, making direct comparisons less fair.
>
> [7] Mladenovic, Andjela, et al. "Online adversarial attacks." arXiv preprint arXiv:2103.02014 (2021).
>
> [8] Chen, Yongqiang, et al. "Understanding and improving graph injection attack by promoting unnoticeability." arXiv preprint arXiv:2202.08057 (2022).

---

> > ### Comment · Reviewer_BqbP · 2025-04-03
> >
> > I thank the authors for additional experiments and discussion. My comments have been partially addressed and I will increase my scores accordingly, acknowledging the limitations of the work.

---

> > > ### Author Response · Authors · 2025-04-03
> > >
> > > Thank you for raising the score and for your time in reviewing and refining our paper. This is a great affirmation of our work. Your comments are very constructive (e.g., considering the unnoticeability, runtime, generalization of our method, etc.), making our paper stronger. We will address your comments in our revision. Thank you again for your valuable time.
> > >
> > > Best,
> > >
> > > Author

---

### Official Review · Reviewer_EZNr · 2025-03-14

**Overall Recommendation:** 3

**Summary:**

The paper introduces MemFreezing, a novel adversarial attack framework designed to disrupt temporal graph neural networks (TGNNs) under realistic constraints where attackers have limited knowledge of future graph changes. The core idea is to strategically freeze node memories in TGNNs, rendering them unresponsive to subsequent updates and propagating these frozen states through neighboring nodes. Experimental results indicate that MemFreezing consistently undermines the performance of TGNN in diverse tasks, providing a more long - lasting adversarial approach when future knowledge is restricted.

**Claims And Evidence:**

The main claims such as persistent degradation of TGNN performance with cross-freezing and future neighbor simulation mechanisms are well supported.
However, one contribution claims that MemFreezing effectively misleads TGNN predictions across diverse datasets and models even in the presence of defenses. I am afraid that the paper does not test specialized defenses designed to counter memory - freezing attacks, such as dynamic memory reset mechanisms. The defensive strategies mentioned in 5.1 are insufficient and incomplete. Other potential defenses regarding the memory-oriented aspects should be summarized more comprehensively with references in appendix C.2.

**Essential References Not Discussed:**

More works in the area of Memory - Based Graph Networks should be cited. [Memory-Based Graph Networks’20]

**Experimental Designs Or Analyses:**

1. The paper mentions that the effectiveness is less significant on JODIE which uses differences between a node's current and its last update time to decay the memory. This may imply that the effectiveness of the method depends on the model architecture. Therefore, has the evaluation covered a sufficient variety of model types for dynamic graphs?  In part 5.1, only four models are compared, and there is no discussion on the differences in model structures. Thus, it is difficult for readers to determine whether the selection of dynamic graph models is comprehensive.

2. In the "Attack Setup" section of 5.1, the attack budgets only include the percentage of attacked nodes, but the magnitude of the Gaussian noise injecting to "fake future neighbors" in MemFreezing is not specified. Are there similar perturbation constants in other attack methods?

**Methods And Evaluation Criteria:**

The main claims such as persistent degradation of TGNN performance with cross-freezing and future neighbor simulation mechanisms are well supported.
However, one contribution claims that MemFreezing effectively misleads TGNN predictions across diverse datasets and models even in the presence of defenses. I am afraid that the paper does not test specialized defenses designed to counter memory - freezing attacks, such as dynamic memory reset mechanisms. The defensive strategies mentioned in 5.1 are insufficient and incomplete. Other potential defenses regarding the memory-oriented aspects should be summarized more comprehensively with references in appendix C.2.

**Other Comments Or Suggestions:**

The white - box attacks employed in the article yield highly remarkable results. Nevertheless, their usability in the real-world context may be constrained to a certain extent. It would be advisable to consider incorporating additional black - box attack methods, such as zero-shot transfer attacks.

**Other Strengths And Weaknesses:**

1. The paper introduces cross-freezing to stabilize node memories but does not provide a formal proof of stability. Can you mathematically demonstrate that the system of mutually reinforcing nodes will remain in a frozen state indefinitely under dynamic graph updates?
2. In the appendix, The black-box evaluation uses surrogate models trained on partial data. How would MemFreezing perform against a truly unknown, closed-source TGNN? The credibility of the paper would be enhanced if the test results for closed - source models could be presented within the article.

**Questions For Authors:**

During simulating the future neighbors of victim nodes, could you please provide the reasons for choosing the two initialization methods(noise nodes and as all - zero nodes)? Is there any relevant previous literature or theoretical basis to support this choice?

**Relation To Broader Scientific Literature:**

Dynamic Graphs are prevalent in real-world scenarios, while Temporal Graph Neural Networks (TGNNs) have become leading solutions for dynamic graph tasks. There is a pressing need to study their robustness towards adversarial attacks. Hence, in real-world cases, the adversarial may attack TGNN without knowing future changes on the graph. To address the challenges, this paper introduce MemFreezing, a novel adversarial attack framework that delivers longlasting and spreading disruptions in TGNNs without requiring post-attack knowledge of the graph.

**Theoretical Claims:**

Part 4.1 claims that node memories in TGNN can remain stable when surrounded by neighbors with similar memories. Furthermore, the ideal frozen states from different nodes are similar.
However, it is not comprehensive enough to draw this conclusion merely from a single experiment with only 100 victim nodes sampled. Is there any support from previous literature or more detailed derivations of theoretical formulas?

---

> ### Author Rebuttal · Authors · 2025-04-01
>
> **We sincerely thank the reviewer for the positive feedback and valuable comments. In response, we clarify the rationale behind the future simulation choices, discuss more block-box attack setups, and clarify the observation on stable states and attack budget. We will also carefully revise the paper following the reviewer’s suggestion.**
>
> ---
>
> ## **Q1. Why simulate a neighbor with noisy and all-zero nodes?**
>
> The reasons for using noisy and all-zero nodes for future simulation are:
>
> - **Noise Nodes:** As indicated by the network homophily principle [1,2] (**as cited in Q3 for reviewer qaAY**) and  Figure 3(d) and Appendix B.16, real-world neighbors often have high memory similarity. Hence, we create neighbors with similar features to the victim node’s current neighbors to simulate potential neighbors that already exist in the graph.
>
>
> - **All-Zero Nodes:** In many TGNNs (e.g., TGN, JODIE, DyRep, etc.), new nodes start with an all-zero memory. Thus, we initialize some future neighbors with zeros to mimic newly added nodes to the graph.
> Together, these strategies capture both the homophilous nature of existing neighbors and the default state of new nodes.
>
> ---
>
> ## **Q2. Zero-shot black-box attack**
>
> We agree that further exploring more black-box constraints is valuable. Hence, we evaluate Memfreezing under the zero-shot transfer attack setup following the ensemble-based approach first proposed in [3]. Specifically, on the Wikipedia dataset, we generate the fake message by jointly optimizing the adversarial message for three models and then evaluate the effectiveness of this unified adversarial message on diverse models. As shown in **Figure R1(anonymous link:https://ibb.co/6pkw68K)**, although it performs worse than the white-box attack, Memfreezing can still effectively perturb model predictions. We also acknowledge that memfreezing is less harmful than cases with more accurate model information.
>
> [3] Liu, Yanpei, et al. "Delving into transferable adversarial examples and black-box attacks." arXiv preprint arXiv:1611.02770 (2016).
>
> ---
>
>
> ## **Q3. Theoretical proof of cross-freezing**
>
> Please refer to **Q3 for reviewer qaAY** for more quantitative demonstrations of similar stable states. We provide the mathematical proof of cross-freezing under the similar stable states as detailed in **Figure R2 (anonymous link: https://ibb.co/XfVksqdR).**
>
> ---
>
>
> ## **Q4. The defensive strategies**
>
> Thank you for your valuable feedback. To our knowledge, no published methods currently detect or mitigate the specific mechanism of node memory freezing in TGNNs.  However, we agree that one could devise a “memory reset” module that resets victim nodes’ memories.
> To explore the potential of randomly and periodically reset node memories, we conduct experiments in which we (a) random reset node memories upon each update or (b) reset memories after each 25 timestamps. The results are shown in **Figure R3 (anonymous link:https://ibb.co/NMrZsjq)**. As one can observe, doing so may jeopardize the models’ clean accuracy with limited effectiveness in defending against Memfreezing. As detailed in **Q2 to reviewer qaAY**, the key challenge is that doing so may overkill those naturally stable nodes, and we also follow up with a potential detecting scheme detailed in that response.
>
> ---
>
>
> ## **Q5. Attack budget in terms of noise magnitude.**
>
> We specify the magnitude of our Gaussian noise in Appendix A.2, where each “fake future neighbor” has a mean of 0 and a standard deviation set to 0.2 times the standard deviation of its corresponding real neighbor’s memory.
> Note that we do **not** inject simulated fake neighbors into the graph; instead, we use their noisy states to generate a single adversarial message per victim node, which matches the theoretical min/max for the clean messages (see Appendix B.10 for a stealthiness analysis).
>
> ---
>
> ## **Q6. Diversity of the models**
>
> We appreciate the reviewer’s suggestion to evaluate MemFreezing on a broader range of TGNN architectures. The four TGNNs represent diverse memory-update mechanisms across different temporal paradigms. Specifically, TGN and DyRep operate event-driven updates, while ROLAND processes discrete snapshots. Moreover, these models employ varied memory schemes: TGN and DyRep use event-based updates, JODIE incorporates a time-decay component, and ROLAND leverages repeated GNN layers or attention over snapshots.
> We acknowledge that Memfreezing mainly targets memory-based TGNNs. As recent studies demonstrated [4], this TGNN family achieves SOTA performance for dynamic graph benchmarks. We will also discuss the memory-based graph network [5] in our revised paper, as suggested by the reviewer.
> [4] Huang, Shenyang, et al. "Temporal graph benchmark for machine learning on temporal graphs." Advances in Neural Information Processing Systems 36 (2023): 2056-2073.
>
> [5] Ahmadi, Amir Hosein Khas. Memory-based graph networks. University of Toronto (Canada), 2020.

---

### Official Review · Reviewer_qaAY · 2025-03-14

**Overall Recommendation:** 4

**Summary:**

The method makes use of a design component (node memory states) of recent temporal graph neural networks to disturb model predictions in future unseen time steps. This is done by selecting high-degree nodes (referred to as root nodes) and their 2 neighbours with the highest node degrees (referred to as support nodes) and adding noise to their node features, so as to achieve "cross freezing" of node memories. Doing so is done mainly at time 0 (but at other times as well in the side experiments).

The flow of the paper is clear, I appreciate the clear writing.

**Claims And Evidence:**

Yes.

**Essential References Not Discussed:**

This is one of my concerns.

In the paper it is mentioned that
"... While several studies have explored the effectiveness of adversarial attacks on dynamic graphs (Lee et al., 2024; Sharma et al., 2022; 2023; Chen et al., 2021), **they often** assume that attackers have complete knowledge of the input graphs at the time of the attack ..."

It seems there have been previous adversarial attack method for dynamic graphs, as mentioned later in the paper:

"... Recently, there have also been a few studies that explored the effectiveness of adversarial attacks on dynamic graphs and TGNNs (Lee et al., 2024; Sharma et al., 2023; 2022; Chen et al., 2021). ...".

The distinctions of the proposed method should clearly be mentioned.

**Experimental Designs Or Analyses:**

Yes.

**Methods And Evaluation Criteria:**

Yes, they totally support the idea.

**Other Comments Or Suggestions:**

Not included in score: typo in Sec. 2.

"... Recent TGNNs focus on CTDGs since they can retain more information than DTDGs’ fixed intervals and more complex (Kazemi et al., 2020). ..."

'and more complex' should be corrected to 'and are more complex'.

**Other Strengths And Weaknesses:**

-

**Questions For Authors:**

- In Sec. 3 under "Attacker’s Capability", in the provided social media example could you describe how the cross freezing of the proposed method would look like in that specific example?
- Since the proposed attack method targets a specific design choice of TGNNs, isn't it easy to detect by, e.g., observing the node memory states?
- It is mentioned that "... in Figure 3(d), the ideal frozen states from different nodes are similar; therefore, it ...".
Does this property hold for any real-world graph in general? and if it's violated, how does it affect the performance of the method?

**Relation To Broader Scientific Literature:**

-

**Theoretical Claims:**

The claims are supported emprically.

---

> ### Author Rebuttal · Authors · 2025-04-01
>
> **We sincerely thank the reviewer for the positive feedback and valuable comments. In response, we exemplify the cross-freezing in a social media case, discuss its potential defenses, and clarify the assumption of similar ideal stable states among nodes. We will also carefully revise the paper following the reviewer’s suggestion.**
>
> ---
>
> ## **Q1. Cross-freezing in social media**
>
> In a social media setting (e.g., Reddit, Facebook), an attacker may want to consistently deliver the content to a specific user (the victim). First, they collect the victim’s profile and its neighbors from public data. Next, for each victim node and its two supportive neighbors, the attacker creates a fake account and injects an adversarial message, such as malicious comments. These fake accounts and comments are later removed, but the TGNN has already recorded the noisy messages. As a result, the updates related to these messages trigger a “cross-freezing” effect that locks the victim’s (and its neighbors’) memory into a stable, noisy state, making them less responsive to real content. Consequently, the victim may consistently receive the same spam content even though their future actions show no interest in those contents.
>
> ---
>
>
> ## **Q2. Detecting Memfreezing**
>
> Detecting a MemFreezing attack by observing node memory is challenging because nodes can naturally exhibit stable updates. For example, using TGN on the Wikipedia dataset, over 70% of node updates show high similarity (although they may not be consistently stable), which can also occur in real-world cases (e.g., an Amazon user with consistent shopping preferences). Thus, it is hard to differentiate an attacked node from naturally stable ones.
>
> However, your insight also hints at a potential detection strategy: deliberately stimulate node changes and then observe how the node’s memory reacts. Concretely, we may check if a node is attacked by Memfreezing in the following two steps:
>
> 1. **Introduce Divergent Neighbors**: Temporarily connect the node to new neighbors with significantly different memory states (e.g., low cosine similarity).
>
>
> 2. **Monitor Update Response**: If the node remains unusually stable after interacting with these distinct neighbors—showing little or no memory shift—it suggests that MemFreezing may be in effect.
>
> By testing for unnaturally persistent memory under a deliberately introduced variation, one can check whether the node is compromised by the MemFreezing attack.
>
> ---
>
> ## **Q3. Observation of similar ideal frozen states**
> Thank you for the insightful question. The similar ideal frozen states widely exist in diverse real-world graphs. In addition to the experiments in Figure 3d, we further characterize the distributions of similarities between neighboring nodes’ stable states across diverse benchmarks and present the results in **Figure 32 of Appendix 16B** in our original paper. The results indicate that the property widely exists in other real-world graphs (i.e., Reddit and Reddit-body). This also aligns with the network homophily theory [1,2], which states that, based on node attributes, similar nodes may be more likely to connect to each other than dissimilar ones.
>
>
> We also investigate the performance under the cases where connected nodes may have divergent features, which potentially give divergent ideal states, in Appendix B16. Specifically, we have victim nodes in the graph connected to nodes with random memories after the attack timestamps. In such cases, the nodes tend to be divergent and pose distinct ideal stable states. The results in **Figure 33 in Appendix B16** indicate that MemFreezing effectively freezes these random neighbors despite resulting in lower similarities.
>
>
> [1] McPherson, M., Smith-Lovin, L., & Cook, J. M. (2001). "Birds of a Feather: Homophily in Social Networks". Annual Review of Sociology. 27:415–444.
>
> [2] Himelboim, I., Sweetser, K. D., Tinkham, S. F., Cameron, K., Danelo, M., & West, K. (2014). Valence-based homophily on Twitter: Network Analysis of Emotions and Political Talk in the 2012 Presidential Election. New Media & Society.

---

### Decision · Program_Chairs · 2025-05-01

**Decision:**

Accept (poster)

**Comment:**

All reviewers positively evaluated the paper and found the core idea of strategically freezing node memories to be interesting.  While the attack is not general and only affects TGNNs with a certain type of architecture, such models are common and often SOTA. The reviewers generally agree that the main claims are well supported. The majority of raised questions and issues have been sufficiently addressed in the rebuttal. One reviewer increased their score.

As Reviewer BqbP notes the paper claims increased practicality due to limited knowledge of the future graphs, but gives full access to the model, i.e. we have a white-box attack. Reviewer EZNr raised a similar question about considering a black-box variant via transfer. The authors provided additional experiments showing that Memfreezing is still partially effectively. Nonetheless, the claims of practicality should  be appropriately "softened" in the final version to avoid over-claiming.

Taking everything into account I recommend acceptance.